# Technology in farming: Unleashing farmers' behavioral intention for the adoption of agriculture 5.0

Nitesh Mishra[1], Nabin Bhandari[2], Tek Maraseni[3,4], Niranjan Devkota[5]*, Ghanashyam Khanal[6], Biswash Bhusal[7], Devid Kumar Basyal[1], Udaya Raj Paudel[1], Ranjana Kumari Danuwar[1]

1 Quest International College, Pokhara University, Gwarko, Lalitpur, Nepal, 2 Agricultural Economics and Rural Sociology, Auburn University, Auburn, Alabama, United States of America, 3 University of Southern Queensland, Toowoomba, Queensland, Australia, 4 Northwest Institute of Eco-Environment and Resources, Lanzhou, China, 5 Patan Multiple Canpus, Tribhuvan University, Patandhoka, Lalitpur, Nepal, 6 College of Forestry, Wildlife and Environment, Auburn University. Auburn, Alabama, United States of America, 7 Department of Applied Economics, John Hopkins University, Baltimore, Maryland, United States of America

* niranjandevkota@gmail.com

**Data Availability Statement:** All relevant data are within the manuscript. It is also uploaded full dataset with the manuscript.

## Abstract

The agriculture sector has undergone a remarkable revolution known as Agriculture 5.0 (Ag 5.0), emphasizing digital technology to boost efficiency and profitability of farm business. However, little is known about farmers' behavioral intension to adopt Ag 5.0. In this study we examine factors influencing farmer's behavioral intension for Agriculture 5.0, identify implementation obstacles and provide managerial solutions to promote Ag 5.0 in Madhesh Province, Nepal, using the Technology Acceptance Model (TAM) and Structural Equation Model (SEM). We tested total of 20 different hypotheses. Primary data were collected from 271 farmers across 9 municipalities in Saptari District, Nepal. The study reveals that technology anxiety [($\beta$ = 0.101, p<0.01); ($\beta$ = 0.188, p<0.01)], self-efficacy [($\beta$ = 0.312, p<0.01, ($\beta$ = 0.170, p<0.05)] and social influence [($\beta$ = 0.411, p<0.01), ($\beta$ = 0.170, p<0.05)] significantly impact the perceived usefulness as well as perceived ease of use, respectively. Individual innovativeness also affects the perceived usefulness ($\beta$ = 0.004, p<0.05) and perceived ease of use ($\beta$ = 0.281, p<0.01). Moreover, the study found that attitude towards using Ag 5.0 is significantly influenced by perceived usefulness ($\beta$ = 0.083, p<0.10) and ease of use ($\beta$ = 0.189, p<0.01), which, in turn, affects the intention to use Ag 5.0 ($\beta$ = 0.858, p<0.01). Farmers perceive training programs, government assistance, and subsidies as helpful in overcoming challenges associated with adopting Ag 5.0. This study provides valuable insights for policymakers, development partners, and farmers' organizations, enabling them to understand the factors influencing the readiness for Ag 5.0 adoption in Nepal.

## 1. Introduction

The possibility of transforming the agriculture sector primarily stems from technological advancements, their dissemination at the community level, and the widespread adoption of

**Funding:** The author received no specific funding for this work.

**Competing interests:** The authors have declared that no competing interests exist.

improved technology by farming communities [1–4]). The adoption of improved practices by farmers is a crucial approach to sustainably intensify the agriculture sector [5–7], which is tasked with feeding an estimated global population of 9.7 billion by 2050. This challenge is compounded by diminishing land and water resources, as well as the impacts of climate change [8–10].

In recent times, scholars have expressed notable concern regarding "Agriculture 5.0 (Ag 5.0)", which aims to enhance productivity, profitability and sustainability of agriculture system [6]. It focuses on the development and utilization of digital smart technologies such as the Internet of Things (IoT), artificial intelligence, machine learning, and data utilization to improve efficiency in the agriculture sector [11]. While automated and cutting-edge farming technologies have been widely used in developed countries [12] their adoption in Nepal has been limited [13,14].

Although agriculture has a long history dating back to human civilization, the development and utilization of high-yielding crop varieties that respond well to pesticides and chemical fertilizers began in the late 1950s [15–17]. The widespread adoption of these improved varieties resulted in an outstanding increase in food-grain production from 1 billion tons in 1960 to 2 billion tons in 2000 [18]. This remarkable surge in food production through substantial genetic improvement of domestic crop varieties is known as the Green Revolution [19,20]. The success of Green Revolution has motivated scientists to shift to the application of information technologies in the agricultural sector to enhance planning decisions and output [21,22], which marked the emergence of Agriculture 4.0. Agriculture 4.0 has given rise to growing interest in Ag 5.0—an innovative approach that leverages AI-based smart technologies and IoT to revolutionize food production and productivity. By employing predictive, detecting, and controlling capabilities, Ag 5.0 focuses on optimizing various farming system aspects, such as real-time evaluation of micro parameters like light, soil, humidity, precipitation, and temperature [23,24].

In Nepal, agriculture plays a dominant role in livelihood and employment generation [25,26] and is considered a cornerstone of economic prosperity [27]. It contributes approximately 23.9 percent to the total GDP in FY2021/2022 and provides employment opportunities for 60.4 percent of the population [28]. Recognizing its importance in the national economy, various policies have been formulated and implemented to guide the agricultural sector in Nepal. One of the most discussed and historic policies is the Agriculture Perspective Plan (Agriculture Perspective Plan) which served as a guiding policy instrument for a 20-year period (1995–2015) to transform the entire national economy [29]. The primary objective of the APP was to alleviate poverty and improve the living standards of the people by achieving accelerated growth in agriculture. This plan aimed to bring overall economic transformation through a technology-based green revolution, with specific package approaches tailored to the Terai, Hills, and Mountain's regions of Nepal.

Taking lessons from the formulation and implementation of the APP, the Government of Nepal has introduced the Agriculture Development Strategy (ADS) for a twenty-year period (2015–2035). This strategy envisions the development of a self-reliant, sustainable, competitive, and inclusive agriculture sector that drives economic growth, improves livelihoods, and ensures food and nutrition security, ultimately leading to food sovereignty [30]. It also emphasizes the utilization of smart technology in agriculture to enhance competitiveness on a global scale [26]. Similarly, another important policy is the National Agriculture Policy (NAP), which focuses on increasing production and productivity, promoting commercialization, and conserving and utilizing natural resources and biodiversity [31]. Additionally, there are commodities and sector specific policies such as fisheries, dairy, agro-forestry, food safety, fertilizer, tea, coffee, irrigation, and biodiversity as well as policies addressing cross-cutting issues like

Information and Communication Technology (ICT), National Science, Technology, and Innovation Policy (NTIP), and climate change [32].

Due to its diverse climate and geographic situation, Nepal has the potential to achieve high and inclusive economic growth by increasing agricultural productivity [33]. Despite this, Nepal has become a net food importer since the early 1980s [34]. The performance of the agriculture sector has been unsatisfactory [35], mainly due to poor policy implementation and inadequate allocation of resources [36]. The average productivity of most crops in Nepal is lower compared to neighboring countries. For instance, in China, the average paddy productivity for the year 2022/2023 is 7.1 tons per hectare, which is remarkably higher than Nepal's average (3.1 tons) [28,37]. Factors such as easy access to irrigation, improved seeds and breeds, chemical fertilizers, pesticides, agricultural loans, advanced farming technologies, and technology know how play a crucial role in increasing production and enhancing productivity in the agriculture sector [38].

In Nepal, the majorities of farmers rely on natural resources and have tendency to adhere to traditional farming practices [39]. Lack of the technological knowledge, skills, and entrepreneurship to transition from traditional farming to a commercial farming has led to existence of vicious circle of poverty among farmers. Realizing the low mechanization of farming, government of Nepal aims to conduct social marketing campaigns emphasizing the advantages of new technologies over traditional forms of cultivation, harvesting etc [30].

Meeting domestic food demand and ensuring food and nutrition security for the population can be achieved by embracing advanced farming technologies [39,40]. Given the decline in farmland, labor shortages, and increased risks from climate change and natural disasters in the agriculture sector, adopting an Ag 5.0 approach is a potential solution to enhance agricultural productivity in Nepal. While there have been various studies on agriculture as general and protected agriculture as specific, there is currently a lack of literature that examines the readiness for and obstacles to adopting agriculture 5.0 among farmers in Nepal. This paper examines the factors influencing readiness for Ag 5.0, identify the obstacles to implement Ag 5.0 0, and provides managerial solutions to promote Ag 5.0 in Madhesh Province, Saptari-Nepal. Furthermore, we believe that identification of farmer's beliefs and opinion towards certain subject matters can provide a grassroots perspective that is essential for the development of effective and realistic policies. In addressing these objectives, we have formulated and tested a total of 20 different hypotheses.

This study provides valuable insights for policymakers, development partners, and farmers' organizations, enabling them to understand the factors influencing the readiness for Ag 5.0 adoption in Nepal and provide them alternative solution to overcome the hurdles associated with the implementation of Ag. 5.0 in Nepal. These findings and solutions could be applicable in many other developing countries with similar seriocomic settings.

## 2. Research methodology

This study outlines a comprehensive research methodology, including the conceptual framework, hypothesis formulation, and research design, detailing the systematic approach for data collection and analysis. It describes the study area and population, the sampling technique, and the sample size to ensure representativeness. The research instruments and data collection methods are specified, ensuring reliability and validity. The study employs Structural Equation Modelling (SEM) to test the relationships between variables and validate the conceptual framework. Each section is designed to provide a clear and structured approach to investigating the research questions, ensuring robust and actionable findings.

## 2.1 Conceptual framework

Several theories have been developed to study factors influencing the readiness, acceptance, and implementation of new technology. The major theories considered in this study for the adoption of Ag 5.0 are the theory of diffusion of innovation [41], unified theory of acceptance and use of technology [42], technological acceptance model [43], theory of reasoned action [44], theory of technological paradigm [45], theory of disruptive innovation [46], and Technological-Organizational-Environmental (TOE) framework. Rogers' theory of diffusion of innovation explains how new ideas or technology spread in a social system, considering characteristics of the innovation, communication channels, time of development, and the social system's impact on technology adoption. It suggests that the adoption curve follows a sigmoid-shaped pattern in society. This theory can help identify the factors influencing the adoption of new technology in the agricultural sector [47,48].

The unified theory of acceptance and use of technology explains the user's adoption of IT, the ease of use of technologies, and their acceptability. It also focuses on utilizing modern tools and techniques to enhance the quality and quantity of agricultural products through the management of real farming data. The technology acceptance model (TAM), as identified by Diop et al. [49], considers perceived ease of use and perceived usefulness as key elements influencing individuals' intention to adopt new technology. This model explores whether the acceptance or rejection of a technology is based on differences in perceived usefulness and ease of use. The theory of reasoned action examines how user ideas and attitudes impact individual performance intentions [50]. The theory of technological paradigm explains the development of technology and innovation, while the technological-organizational-environment framework explores how businesses adopt and introduce technological innovations based on technology, organizational, and environmental factors. These models collectively analyze the various factors that affect the process of technology adoption and implementation.

Among the mentioned theories, TAM is suitable for researching Agriculture 5.0 as it identifies factors influencing the adoption of modern technologies in agriculture. Perceived usefulness (PU) refers to the belief that a system improves job performance, indicating its value. Perceived ease of use (PEOU) refers to the belief that a system is user-friendly and effortless. Complex technologies are less likely to be adopted by the target group. Fig 1 illustrates the basic idea of Davis et al.'s [43] TAM. In relation to agriculture, TAM can describe the acceptance of smart technologies and their associated benefits.

The TAM model has been applied in various studies. Castiblanco et al. [51] measured the acceptance of an e-learning tool for EU farmers using TAM. Salimi et al. [52] analyzed the factors influencing the adoption of agricultural automation using TAM. Similarly, Rezaei-Moghaddam et al. [53] evaluated the perception of Iranian agricultural specialists regarding grid soil sampling technology using external variables. Piot-lepetit et al. [54] studied IT adoption in agriculture using the integrated TAM-TOE model. Rezaei-Moghaddam & Salehi [55] investigated the intention and attitude toward precision agriculture technologies, considering external factors such as confidence, traceability, and observability.

Based on the examination of various variables, including AI technology anxiety, AI technology self-efficacy, Individual Innovativeness, facilitating conditions, social influence, perceived usefulness, perceived ease of use, and attitude towards using Ag 5.0, we constructed a conceptual framework to assess farmers' readiness for adopting Agriculture 5.0 in Madhesh Province. The conceptual framework (Fig 2) integrates Castiblanco et al. [51] conceptual model and includes five external factors, along with perceived ease of use (PEU) and perceived usefulness (PU) as mediating variables, while the response variable encompasses attitude towards using Ag 5.0 and behavioral intention.

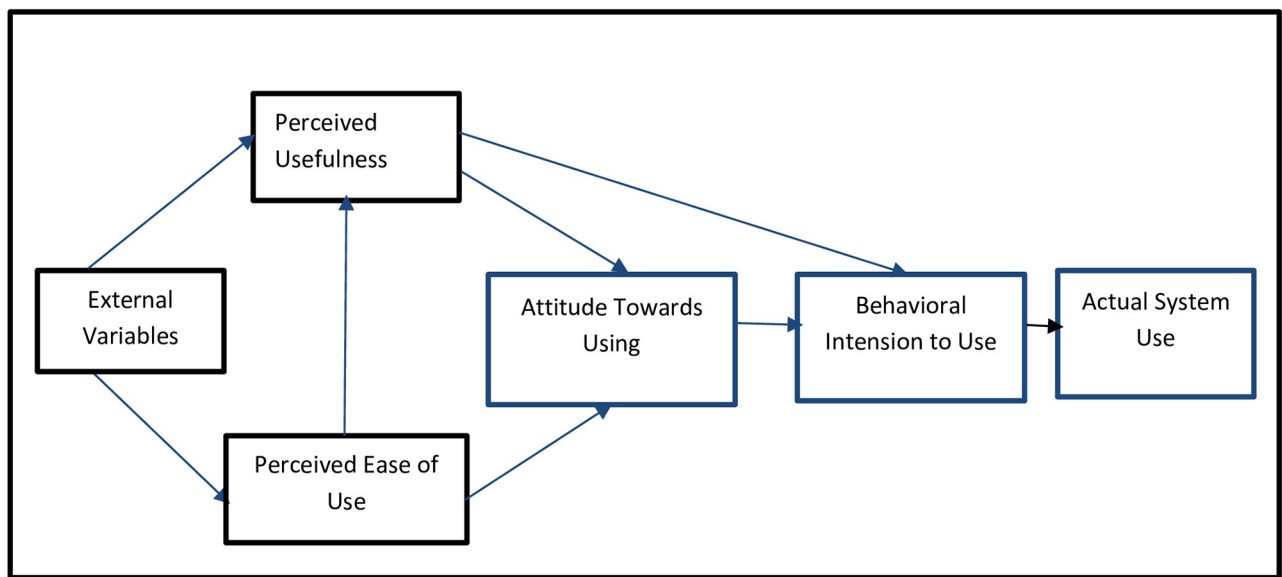

**Fig 1. Technology acceptance model.**

**Hypothesis formulation.** *Technology anxiety and perceived usefulness and technology anxiety perceived ease of use.* Technology Anxiety refers to individuals' apprehension or fear when using technology [56]. It is associated with negative emotions and stress resulting from challenges and negative beliefs about technology [57]. Comfort with technology has a positive

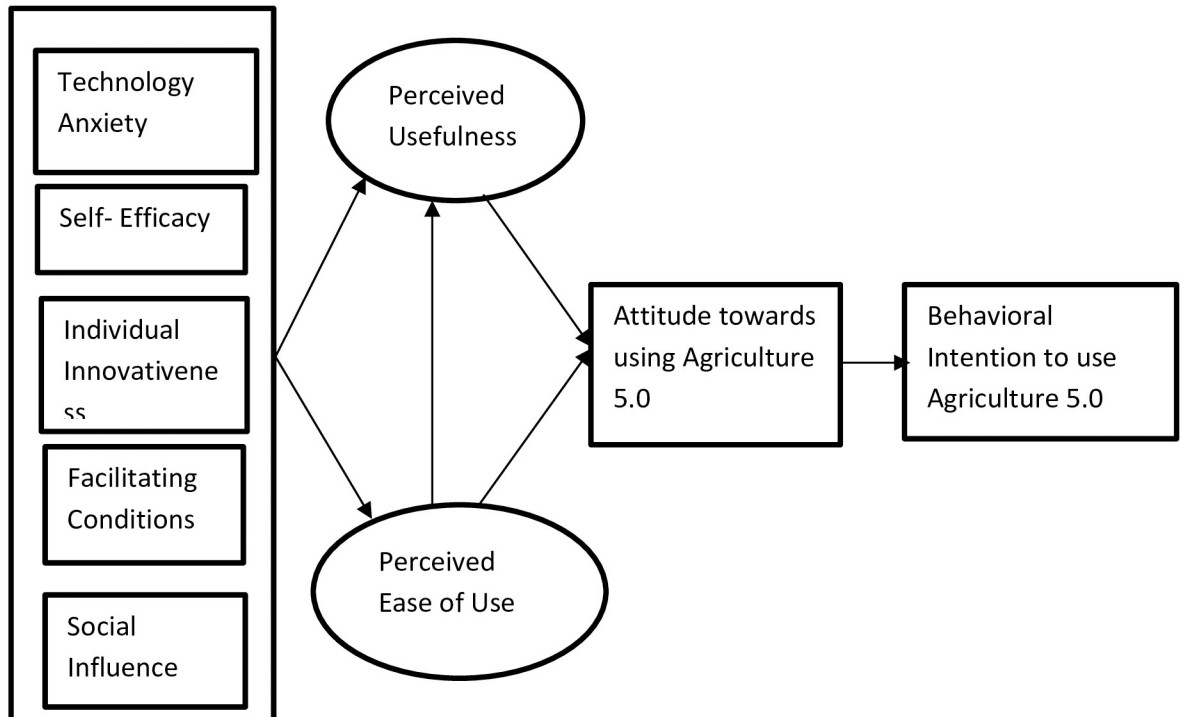

**Fig 2. Conceptual framework for assessing farmers' behavioral intension to adopt agriculture 5.0 in Madesh Province.**

impact on Perceived Usefulness, leading to positive outcomes, enhanced performance, and enjoyment. Conversely, high anxiety levels are linked to lower productivity, reduced performance, and discomfort, often resulting in inflexible behavior [58]. User-friendly technology reduces anxiety levels, and with experience, users accurately assess task effort and identify enjoyable system features, minimizing anxiety's impact on perceived ease of use [59]. Considering these findings, the adoption of Ag 5.0 technology may initially cause anxiety among farmers, given the potential stress and discomfort associated with its use, especially for first-time users.

**H1**: *Technology Anxiety has significant impact on Perceived Usefulness to adopt Ag 5.0*

**H2**: *Technology Anxiety has significant impact on Perceived Ease of Use to adopt Ag 5.0*

*Self-Efficacy and Perceived Usefulness and Self-Efficacy and Perceived Ease of Use.* Self-efficacy refers to users' confidence and ability to effectively use technology [60,61]. It includes the skills and knowledge necessary to accomplish tasks with technology [61]. In the context of technology, self-efficacy involves confidence in technology-related awareness and comfort in using new technology [60]. Self-efficacy also plays a crucial role in determining the perceived usefulness (PU) and perceived ease of use (PEOU) of new technology [62]. Certain skills and knowledge give users confidence in using technology, making it appear easy to use, while understanding the goals and design principles behind the technology makes it seem useful [63]. Farmers can develop confidence and skills to effectively use Ag 5.0 technologies, influencing both perceived usefulness and perceived ease of use. Thus, self-efficacy has a positive impact on both [60,62].

**H3**: *Self-efficacy has significant impact on Perceived Usefulness and Perceived ease of use. to adopt Ag 5.0*

**H4**: *Self-efficacy has significant impact on Perceived ease of use to adopt Ag 5.0*

*Individual innovativeness and perceived usefulness and perceived ease of use.* Innovativeness refers to individuals' willingness to experiment with new technologies [64]. It involves attitudes towards innovation and the adoption rate of new technologies. Innovativeness is associated with early acceptance of new ideas, leading to positive beliefs about using technology [65]. There is a positive relationship between Individual Innovativeness and Perceived Usefulness, as innovative individuals are more likely to try new technologies. Similarly, Individual Innovativeness is positively related to Perceived Ease of Use, as innovative individuals are eager to adopt and use new technology [66]. Based on this research, it can be concluded that farmers with high innovativeness are more likely to adopt Ag 5.0 technologies. Therefore, the hypothesis is as follows:

**H5**: *Individual Innovativeness has significant impact on Perceived Usefulness to adopt Ag 5.0*

**H6**: *Individual Innovativeness has significant impact on perceived ease of use to adopt Ag 5.0*

*Facilitating conditions and Perceived Usefulness and facilitating conditions and Perceived Ease of Use.* Facilitating conditions refer to individuals' perception of the availability of technological and organizational resources that support system use [67]. This includes external resources like time, money, and effort, as well as necessary technology resources such as AI technologies and machines that make performing a specific behavior easier [68]. The presence of facilitating conditions impacts both perceived usefulness (PU) and perceived ease of use (PEOU), as higher levels of technical support contribute to more positive attitudes and a greater intention to use AI technology [69]. Based on this evidence, it can be concluded that if

farmers have access to the necessary resources for utilizing Ag 5.0 technology, they will perceive the technology as useful and easy to use, ultimately enhancing their performance. Therefore, the proposed hypothesis is as follows:

**H7**: *Facilitating conditions has significant impact on Perceived Usefulness*. to adopt Ag 5.0

**H8**: *Facilitating conditions has significant impact on perceived ease of use*. to adopt Ag 5.0

*Social Influence and Perceived Usefulness and Social Influence and Perceived Ease of Use*. The opinions of others, rather than personal convictions, can influence people's acceptance of technology [70]. Social pressure plays a significant role in motivating the adoption of new technology and influencing behavior during the adoption process [71]. The influence of others on technology acceptance is known as social influence [72]. Social influence impacts perceived usefulness (PU) and perceived ease of use (PEOU) because when individuals observe others using technology and perceive its benefits and ease of use, they become more willing to adopt and use it, leading to increased present and future usage [73]. Based on this research, it can be concluded that influential individuals with technology experience can motivate farmers to adopt and use Ag 5.0 technologies. Therefore, the proposed hypothesis is as follows:

**H9**: *Social Influence has significant impact on Perceived Usefulness and perceived ease of use*. to adopt Ag 5.0

**H10**: *Social Influence has significant impact on perceived ease of use*. to adopt Ag 5.0

*Perceived Ease of Use and Perceived Usefulness*. Others' opinions, rather than personal convictions, can influence technology acceptance [70]. Social pressure motivates technology adoption and influences behavior during the adoption process [71]. This influence is called social influence [72] and affects perceived usefulness (PU) and perceived ease of use (PEOU). When individuals see others using technology and perceiving its benefits and ease of use, they are more likely to adopt and use it, increasing present and future usage [73]. Based on this research, influential individuals with technology experience can encourage farmers to adopt and use Agriculture 5.0 technologies. Therefore, the proposed hypothesis is as follows:

**H11**: *Perceived Ease of Use has significant impact on Perceived Usefulness*. to adopt Ag 5.0

*Perceived Usefulness and Attitude towards Using Agriculture 5.0 and Perceived Ease of Use and Attitude towards Using Agriculture 5.0*. Attitude refers to a person's inclination and personal experience with a behavior [74]. Perceived usefulness (PU) reflects the belief that using a specific technology enhances task performance, while perceived ease of use (PEOU) relates to the perception of technology being straightforward and comprehensible [43,57]. When individuals perceive technology as useful, easy to use, and compatible with their values and lifestyle, they develop positive attitudes towards its adoption [75]. Based on research evidence, it can be concluded that farmers will have positive attitudes towards using Ag 5.0 technology if it is user-friendly, useful, and improves performance. Therefore, the proposed hypothesis is as follows:

**H12**: *Perceived Usefulness and perceived ease of use have significant impact on Attitude towards Using Agriculture 5.0*

**H13**: *Perceived ease of use has significant impact on Attitude towards Using Agriculture 5.0*

*Attitude towards Using Ag 5.0 and Behavioral Intention to Use Agriculture 5.0*. Behavioral intention refers to an individual's motivation to engage in a specific behavior [76], indicating the likelihood of technology adoption [77]. Attitude represents an individual's evaluation of a

behavior as positive or negative [74]. Thus, behavioral intention is directly influenced by attitude towards technology, where a positive attitude leads to a positive intention to adopt and vice versa 78]. Based on this evidence, it can be concluded that farmers' attitude towards using Agriculture 5.0 technology significantly impacts their behavioral intention. If farmers have a positive attitude towards Agriculture 5.0, their intention to use the technology will also be positive, and vice versa. Therefore, the proposed hypothesis is as follows:

**H14**: *Attitude towards using Agriculture 5.0 has significant impact on Behavioral Intention to use Agriculture 5.0.*

*Perceived Ease of Use (PEOU) as a Mediator.* Perceived ease of use refers to how effortlessly a person perceives using technology, requiring minimal effort. It affects technology adoption by influencing the level of work involved in learning and using technology [79,80]. When users find technology user-friendly and convenient, they believe it enhances their performance. Based on this understanding, it is hypothesized that perceived ease of use significantly impacts technology anxiety, self-efficacy, individual innovativeness, facilitating conditions, social influence, and perceived usefulness. Therefore, the proposed hypothesis is as follows:

**H15**: *Perceived Ease of Use (PEOU) mediates the relationship between Technology Anxiety and Perceived Usefulness (PU)*

**H16**: *Perceived Ease of Use (PEOU) mediates the relationship between Self-Efficacy and Perceived Usefulness (PU)*

**H17**: *Perceived Ease of Use (PEOU) mediates the relationship between Individual Innovativeness and Perceived Usefulness (PU)*

**H18**: *Perceived Ease of Use (PEOU) mediates the relationship between Facilitating Conditions and Perceived Usefulness (PU)*

**H19**: *Perceived Ease of Use (PEOU) mediates the relationship between Social Influence and Perceived Usefulness (PU)*

*Perceived Usefulness as a mediator.* Perceived usefulness refers to how much a person believes that using a specific technology will improve their job performance. A high perceived usefulness score indicates a positive connection between system usage and achieving better outcomes. It demonstrates that the system is viewed as a valuable tool for completing tasks and encourages technology adoption [43,81,82]. When individuals perceive a technology as advantageous and beneficial for their activities, they develop a positive attitude towards using it. Therefore, perceived usefulness significantly impacts both perceived ease of use and attitude towards using Ag 5.0. Based on this, the proposed hypothesis is as follows:

**H20**: *Perceived Usefulness mediates the relationship between Perceived ease of use and Attitude towards using Agriculture 5.0.*

## 2.2 Research design

Explanatory research is utilized to address the research questions, facilitating the development, extension, and testing of theories [83]. According to Kivunja and Kuyini [84], explanatory research focuses on understanding the relationship between cause and effect and investigates into the reasons and mechanisms behind specific phenomena. Additionally, Sutrisna [85] suggests that explanatory research is suitable when the aim is to identify and document correlations among different aspects of the studied event. Given that our research aims to investigate

the impact of a chosen variable on farmers' readiness to adopt Ag 5.0, this article will be valuable insights to augment the current literature of technology adoption in in the context of Nepal.

**Study area and population.** The study was conducted in Madhesh Province Nepal. Madhesh Province is located in the Terai region of Nepal, bordering Koshi Province to the east, Bagmati Province to the north, and India's Bihar state to the south (Fig 3). Furthermore, this province is recognized as one of the prominent food bowls in Nepal and holds noteworthy popularity in terms of agricultural production. The total land under agriculture is 585,008 hectares (16.44% of total area of province), including vacant areas of 14,065 hectares (0.97%), irrigated areas of 357,936 hectares (25.71%), forested areas of 247,278 hectares (3.85%)., and areas covered by rivers and ponds totaling 49,470 hectares (13.99%). Grazing land covers 2.20% of the total land area. The farmers of Sapatari district were selected to obtain the field-level data. We select this district because this district is an agricultural hub and is characterized as high productive district in context of agriculture production in Madehsh Province [86]. Furthermore, most of the population of this district is engaged in agriculture as their primary occupation [87].

**Sampling technique and sample size.** The study applied the purposive sampling technique, in which respondents were chosen at purpose to represent farmers from 9 municipalities in the Saptari district to measure farmers' readiness for the adoption of Agriculture 5.0. Paudel & Devkota [88] also highlighted that purposive sampling is particularly advantageous when researchers need to quickly access a target sample and proportionality is not a primary concern. Given that the sample in this study can only be logically considered representative of the population, we sought expert opinions during the methodology development phase. Their insights guided us in selecting our sample using a nonrandom approach. Hence, the study district was purposively selected after discussions with staff from agricultural departments and

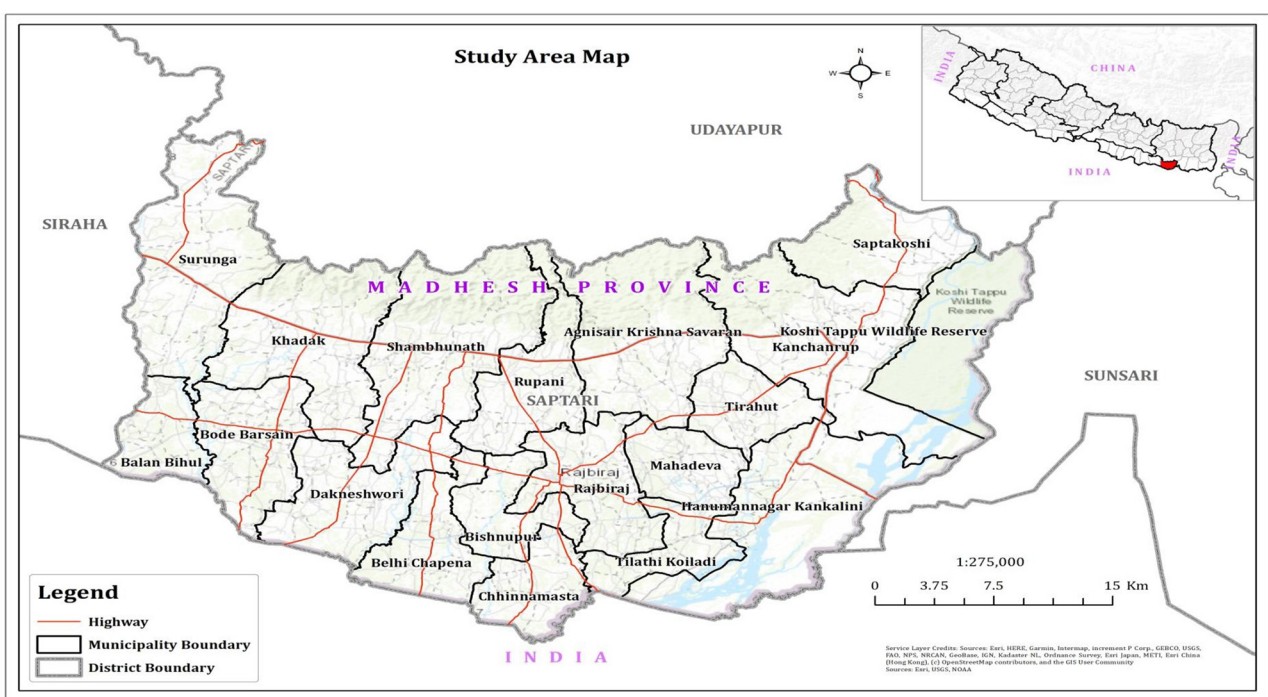

**Fig 3. Study area.**

experts. There are several reasons for selecting this district (agricultural hub in Nepal and many pilot projects are ongoing, etc.). After selecting this district, we delved deeper and found that there are 67,058 households engaged in the agricultural sector, which represents our population.

The appropriate sample size is calculated by using the following formula suggested by Cochran [89]:

$$n = (N * X)/(X + N - 1),$$

Where, N is the sample size required for the study, N (67058) is total number of households engaged in agriculture sector, which represents population, X is represented by $\frac{(Z_{\alpha/_2})^2 * P * (1 - P)}{\varepsilon^2}$, where $Z_{\alpha/_2}$ is the critical value of the distribution, here we assume that critical value to be 1.96 as we consider 95% confidence level at α = 0.05. ε is margin of error (0.05), and P is prevalence proportions. From the above procedure the sample size (n) is calculated to be 246 and adding 5% as non-respondent (12) error our total sample size taken for the study is 258. These sample households were randomly selected using a random table. The survey was conducted from 1st August to 22nd August, 2022.

Given that all participants are small holder agriculture farmers with similar socioeconomic conditions, we firmly believe that our sample size accurately represents the entire agricultural population within these municipalities.

**Research instruments and data collection.** A structured questionnaire, with closed ended questions, was employed to gather information on the respondents' readiness for Agriculture 5.0. A pre-testing of questionnaire survey of 14 respondents (i.e., farmers) was done after the questionnaire was added to KOBO toolbox, to to determine whether the questionnaire makes sense, is workable, and what refinements are necessary to improve its clarity. The questionnaire was refined with a few minor changes as per their suggestions during pre-testing phase. The survey, focusing on human participants specifically farmers from the Saptari districts of Nepal, obtained ethical approval from the Quest Institution Review Committee (QRIC)—registered under number 120 on July 10, 2022. Quest Institutional Review Committee (QIRC), similar to Institutional Review Board (IRB) at universities, at Quest International College with the aim to ensure the ethical conduct of research involving human participants. Additionally, informed consent was obtained from all study participants during the survey, collected through structured interviews. The final data were collected in the month of February and March of 2022. The collected data was finalized and managed using Microsoft Excel, and final inferential data analysis was done using SPSS-AMOS software and SMART PLS 4.0. Table 1 shows the construct and items undertaken for the study.

**Structural Equation Modelling (SEM).** In this research, Structural Equation Modeling (SEM), a second-generation statistical analysis tool, is utilized to investigate the hypotheses formulated in section 2. SEM enables the expression of relationships between variables through a series of single and multiple regression equations, allowing for the modeling of links between explanatory variables and determinant factors. It facilitates the construction of unobserved Latent Variables (LV), model errors, and the evaluation of hypotheses based on quantitative understanding [90]. The usefulness of SEM lies in its ability to specify the system of relationships, measure latent variables using observable indicators, and explore linear causal links among variables while accounting for measurement error. This makes it like, but more effective than, Ordinary Least Square regression analysis [91,92].

The SEM generally consists of two parts i.e., the measurement models and structural equation model.

**Table 1. Variables and their definitions.**

| Construct | Variable ID | Observed Variables | Explanation |
|---|---|---|---|
| Technology Anxiety[1] | TA_1 | Intimidating | Use of technology Frightening/ threatening |
| | TA_2* | Uncomfortable | Uncomfortable in use of technology |
| | TA_3 | Stress | Use of technology stressful |
| | TA_4* | Hesitate | Hesitate to use the system |
| | TA_5 | Apprehensive | Apprehensive about using |
| Self-efficacy[1] | SE_1 | Confident | Feels confident to use the technology |
| | SE_2 | Skill | Skills to accomplish task |
| | SE_3* | Knowledge | Knowledge to accomplish task |
| | SE_4* | Overcoming Obstacles | Overcome obstacles to accomplish task by technology |
| | SE_5 | Belief | Believe to accomplish task |
| Individual Innovativeness[2] | INI_1 | Innovative | Adaptation of innovation |
| | INI_2* | Experiment | Experiment with new learning |
| | INI_3 | Willingness | Ready to adopt the innovations |
| | INI_4* | Openness | Open to accept and the innovation. |
| | INI_5 | Enjoy | Enjoy trying new ideas and innovations. |
| Facilitating Conditions[1] | FC_1 | Guidance | Helpful guidance in performing tasks |
| | FC_2 | Assistance | Available for assistance with system difficulties |
| | FC_3* | Resources | Resources necessary to use the system |
| | FC_4* | Compatible | Not compatible with other systems |
| | FC_5 | Accessible | Easily accessible and understandable |
| Social Influence[1] | SI_1 | Influence Behavior | Influencing the behavior to use the technology. |
| | SI_2* | Encourage | Encourages to use technology |
| | SI_3 | Proportion of coworkers | Proportion of coworker's use of technology |
| | SI_4* | Status Symbol | Perceived to enhance image and status |
| | SI_5 | Supportiveness | Supports the use of technology |
| Perceived Usefulness | PU_1 | Usefulness | Technology is useful for farmers. |
| | PU_2* | Improve Performance | Improve performance of farmers. |
| | PU_3 | Productivity | Increase in productivity |
| | PU_4 | Easiness | Use of technology will make the job easy |
| | PU_5* | Effectiveness | Enhance effectiveness in accomplishment of task |
| Perceived Ease of Use[2] | PEOU_1 | Easy to use | Easy to use technology |
| | PEOU_2 | Clarity & Understandable | Clear and understandable of technology |
| | PEOU_3 | Flexible | Flexible to use |
| | PEOU_4* | Easy to Operate | Easy to operate by farmers |
| | PEOU_5* | Mental effort | Less requirement of mental effort |
| Attitude towards Using Agriculture 5.0[1] | AU_1 | Desirability | Desirable and attractive of technology |
| | AU_2* | Positivity | Positive feelings to use |
| | AU_3 | Goodness | Good and attractive to use |
| | AU_4* | Level of Enjoy | Enjoyable in using the technology |
| | AU_5 | Pleasant | Pleasant to use the technology |
| Behavioral Intention to use Agriculture 5.0[2] | BI_1 | Advantageous | Advantageous to use technology |
| | BI_2* | Favor of Using | Favor in using the technology |
| | BI_3 | Continue to Use | Frequently to use of technology |
| | BI_4 | Recommend | Recommend to others |
| | BI_5 | Beneficial | Beneficial in use the technology |

Note: 1 = Venkatesh and Bala [59] 2 = Castiblanco et al. [51] and '*' items were discarded during data analysis while performing Exploratory Factor Analysis (EFA) and Confirmatory Factor Analysis (CFA) as their factor loading is less 0.50.

According to Pillai and Sivathanu [93], the measurement models can be specified as;

$$y = \Lambda y\, \eta + \varepsilon \tag{1}$$

$$x = \Lambda x\, \xi + \delta \tag{2}$$

And, the structural equation model is specified as:

$$\eta = \alpha + \beta\eta + \Gamma\xi + \zeta$$

Where y = outcome variables, x = input variables, $\Lambda$y = latent variables (observed response variables), $\Lambda$x = latent variables (observed response variables), $\varepsilon$ and $\delta$ are error of Eqs (1) and (2) respectively. $\eta$ = latent variables (unobserved response variables), $\xi$ = latent variables (unobserved response variables) and $\alpha$ = vector of intercepts and $\beta$ = matrix of co-efficient.

## 3 Results and discussions

The section presents a comprehensive analysis of the research findings. It begins with a descriptive analysis and descriptive statistics to summarize the data. It then explores the challenges of Ag 5.0 and provides managerial solutions to overcome these challenges. Inferential statistics are used to draw conclusions from the data. Confirmatory Factor Analysis (CFA) is conducted to assess the measurement model, including tests for convergent and discriminant validity. The section also includes a test of the hypotheses and mediational analysis to understand the relationships between variables and the underlying mechanisms driving these relationships.

### 3.1 Descriptive statistics

The socio-demographic results indicate that most households are male headed (94.46%), while only small percentages are female (5.54%) headed in our survey. The age group of 41–50 represents the largest proportion of respondents (29.15%). Most of the respondents are married (94.1%), and a notable portion of farmers are illiterate (48%). Around 28% have completed secondary education, and only 7% have finished higher secondary education (see Table 2). The study reveals that male headed farmers are more likely to adopt modern farm technologies compared to female headed farmers. However, some respondents, despite being illiterate, have extensive experience, which contributes to their understanding of new technologies and willingness to take risks associated with advanced farming techniques.

Regarding training, the majority of farmers (91.88%) have not received any kind of training, while only 8.12% have received training from different organizations. Among those who received training, over 63% have received the training twice. Among total respondents 7.75% received training from NGOs/INGOs, and 6.64% received it from governmental organizations and 0.74% of the respondents receive training from different other organizations.

The socio-demographic study provides valuable insights for the government, development partners, and farmer organizations. Firstly, it highlights the need to empower women and enhance their entrepreneurial capacity to increase their participation in economic decision-making within households [7]. Secondly, the high percentage of illiteracy among respondents emphasizes the need for targeted education policies to encourage formal education among farming households' children [6]. Additionally, the low coverage of agricultural extension services calls for coordinated efforts between the government, development partners, private sector, and farmer organizations to expand the reach of these services. Furthermore, frequent training programs are justified to keep farmers updated on emerging technologies as they evolve and improve over time.

**Table 2. Socio-demographic profile of respondents.**

| Tittle | Category | Number | Percentage (%) |
|---|---|---|---|
| Gender | Male | 256 | 94.46% |
| | Female | 15 | 5.54% |
| Age | 15–20 Years | 3 | 1.11% |
| | 21–30 Years | 32 | 11.81% |
| | 31–40 Years | 60 | 22.14% |
| | 41–50 Years | 79 | 29.15% |
| | 51–60 Years | 74 | 27.31% |
| | 61–70 Years | 21 | 7.75% |
| | 70 & above | 2 | 0.74% |
| Marital Status | Married | 255 | 94.1% |
| | Unmarried | 16 | 5.9% |
| Education Level | Illiterate | 129 | 47.6% |
| | Up to Secondary level | 76 | 28.04% |
| | Higher Secondary | 41 | 15.13% |
| | Bachelor | 20 | 7.38% |
| | Master | 5 | 1.85% |
| Have you taken farming training | Yes | 22 | 8.12% |
| | No | 249 | 91.88% |
| Farming Training | Up to 2 training | 14 | 5.17% |
| | 3–5 training | 7 | 2.58% |
| | More than 5 training | 1 | 0.37% |
| Farming training Provided by | NGOs/INGOs | 21 | 7.75% |
| | From Government | 18 | 6.64% |
| | Others | 2 | 0.74% |

## 3.3 Challenges of Ag 5.0

The results indicated that only 4.06% of the participants reported no barriers in adopting the new technology. The survey report identified several challenges associated with agriculture, including lack of awareness (71.59%), lack of knowledge and training (67.53%), high cost leading to unaffordability (61.62%), telecommunication infrastructure issues (52.4%), maintenance and repair issues (49.08%), inadequate infrastructure and investment (38.38%), small farm size and land fragmentation (36.9%), and climate change (18.08%) (see Fig 4).

Additionally, respondents were asked if these challenges discourage them from adopting Ag 5.0. Many of the respondents (95.2%) believe that challenges discourage them from adopting Ag 5.0.

## 3.4 Managerial solution to overcome challenges of Ag 5.0

The study examined whether farmers have an optimistic view of the solutions to overcome the challenges discussed earlier. The findings indicate that farmers perceive these challenges as manageable. The number of respondents who provided specific solution strategies is shown in Fig 5. This was a multiple-choice question, allowing respondents to select more than one answer.

## 3.5 Inferential statistics

The normality test of the dataset use for the analysis reveals that the kurtosis value ranges from -1.473 to +3.472 (i.e., between -4 to +4) and its skewness value ranges from -1.443 to -0.196

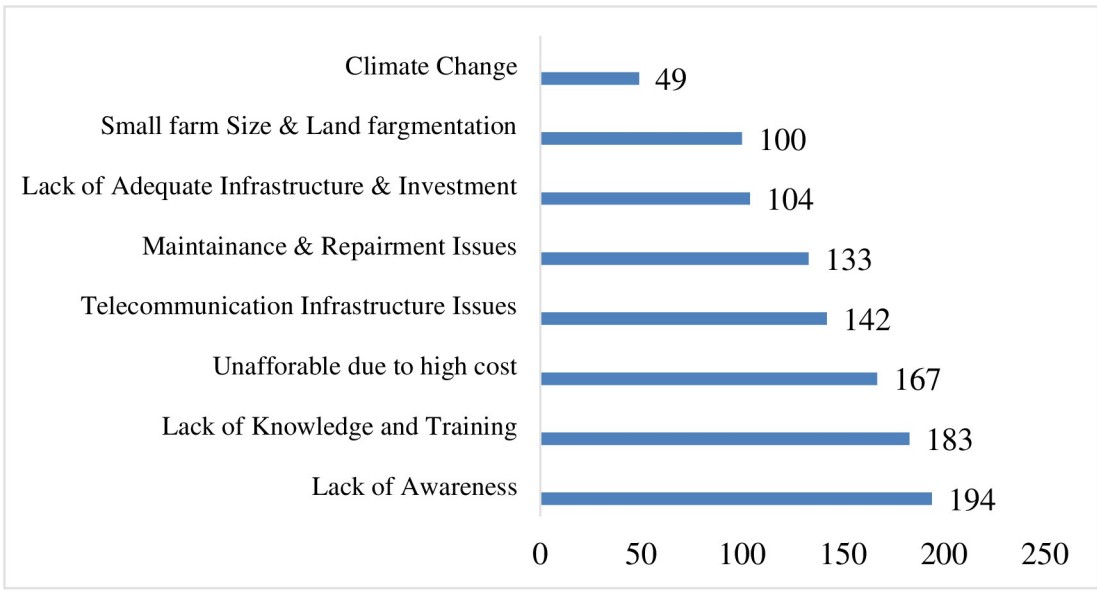

**Fig 4. Farmers response towards the challenges related to Agriculture 5.0 in Madhesh Province (n = 271).**

(between -2 to +2). These findings suggest that the dataset used in the analysis does not exhibit any normality issues. This aligns with the criteria proposed by Black et al. [94], Bryne [95] and Brown [96] who assert that data is considered normal when kurtosis falls within the range of -7 to +7 and skewness is between -2 to +2.

**Confirmatory Factor Analysis (CFA).** We employed Confirmatory Factor Analysis (CFA) to assess and confirm the pre-defined hypothesis concerning the underlying structure

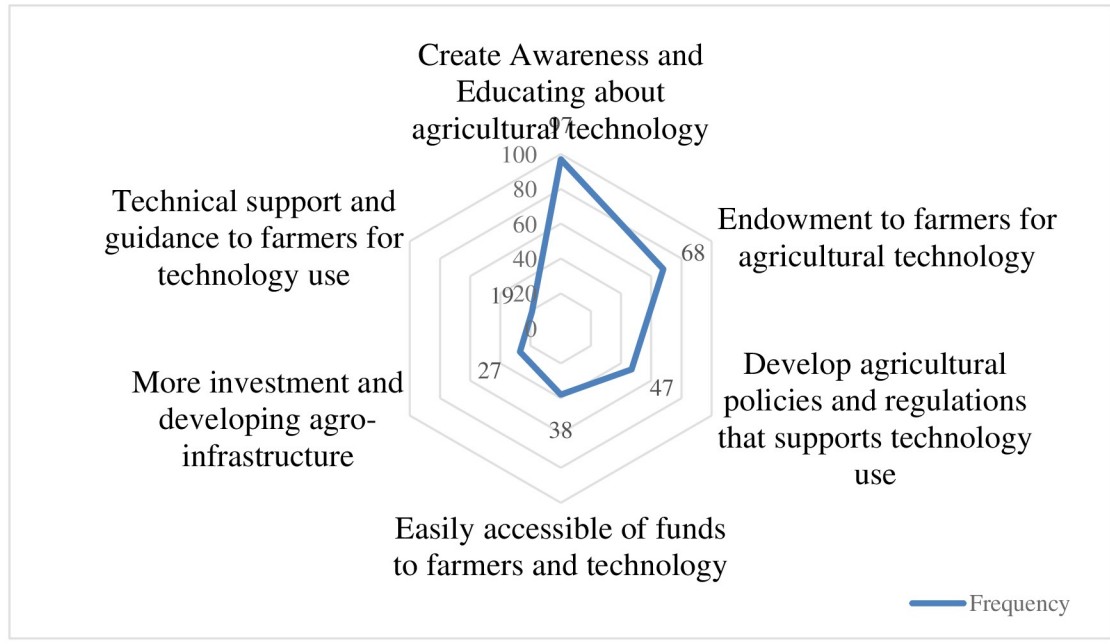

**Fig 5. Managerial solution to overcome challenges related to Ag 5.0.**

of the observed variables. Through CFA, we aimed to evaluate the discriminant validity and reliability of the constructs under investigation. In confirmatory factor analysis, we utilized the three fit indexes to assess the goodness of fit using SEM. A comparative fit index (CFI) values of 0.9 or higher, and root mean square residual (RMR) and root mean square error of approximation (RMSEA) values of 0.1 or lower, suggest a good fit [96,97].

The analysis reveals that the CMIN/DF (chi-square statistics to degrees of freedom) is 2.168 ($<$5), indicating that our model fits the data [98]. The comparative fit index (CFI) is 0.923, which exceeds the threshold ($>$0.9). The root mean square error is 0.034, below the critical value ($<$0.08). These statistics indicate that the measurement models are satisfactory.

**Convergent validity:** Convergent validity measures the level of consistency among multiple items in assessing a single construct. Factor loading, composite reliability (CR), and average variance extracted (AVE) are three indicators used to assess convergent validity. The recommended thresholds for convergent validity are AVE $>$ 0.5 and CR $>$ 0.7 [99]. Additionally, the CR values should be higher than the corresponding AVE values. Table 3 demonstrates that the measurement model surpasses the recommended values.

**Discriminant Validity:** Discriminant validity refers to the ability of predictors in the model to differentiate between constructs and assesses how effectively items measure different concepts across constructs [99,100]. Discriminant validity was evaluated using the Fornell and

**Table 3. Items loading, composite reliability and average variance.**

| Constructs | Indicators | Loadings | Cronbach's Alpha | Compositive Reliability | Average Variance Explained |
|---|---|---|---|---|---|
| Technology Anxiety | TA_1 | 0.891 | 0.944 | 0.947 | 0.856 |
| | TA_3 | 0.829 | | | |
| | TA_5 | 0.920 | | | |
| Self-efficacy | SE_1 | 0.748 | 0.852 | 0.858 | 0.669 |
| | SE_2 | 0.820 | | | |
| | SE_5 | 0.855 | | | |
| Individual Innovativeness | INI_1 | 0.865 | 0.886 | 0.886 | 0.722 |
| | INI_3 | 0.804 | | | |
| | INI_5 | 0.825 | | | |
| Facilitating Conditions | FC_1 | 0.842 | 0.866 | 0.886 | 0.683 |
| | FC_2 | 0.813 | | | |
| | FC_5 | 0.799 | | | |
| Social Influence | SI_1 | 0.758 | 0.824 | 0.824 | 0.609 |
| | SI_3 | 0.757 | | | |
| | SI_5 | 0.766 | | | |
| Perceived Usefulness | PU_1 | 0.827 | 0.842 | 0.844 | 0.645 |
| | PU_3 | 0.799 | | | |
| | PU_4 | 0.750 | | | |
| Perceived ease of use | PEOU_1 | 0.881 | 0.916 | 0.916 | 0.785 |
| | PEOU_2 | 0.861 | | | |
| | PEOU_3 | 0.871 | | | |
| Attitude towards using | AU_1 | 0.705 | 0.770 | 0.773 | 0.534 |
| | AU_3 | 0.748 | | | |
| | AU_5 | 0.698 | | | |
| Behavioral intention to use | BI_1 | 0.882 | 0.954 | 0.955 | 0.840 |
| | BI_3 | 0.887 | | | |
| | BI_4 | 0.885 | | | |
| | BI_5 | 0.889 | | | |

**Table 4. Discriminant validity (Fornel-Lacker method).**

|  | AU | TA | SE | INI | FC | SI | PU | PEOU | BI |
|---|---|---|---|---|---|---|---|---|---|
| **AU** | **0.731** |  |  |  |  |  |  |  |  |
| **TA** | 0.367 | **0.925** |  |  |  |  |  |  |  |
| **SE** | 0.094 | -0.066 | **0.818** |  |  |  |  |  |  |
| **INI** | 0.442 | 0.320 | 0.100 | **0.850** |  |  |  |  |  |
| **FC** | -0.012 | -0.296 | 0.363 | -0.131 | **0.826** |  |  |  |  |
| **SI** | 0.523 | 0.304 | 0.268 | 0.479 | -0.146 | **0.781** |  |  |  |
| **PU** | 0.171 | 0.269 | 0.362 | 0.217 | -0.168 | 0.466 | **0.803** |  |  |
| **PEOU** | 0.405 | 0.419 | 0.175 | 0.415 | -0.173 | 0.409 | 0.125 | **0.886** |  |
| **BI** | 0.478 | 0.043 | -0.081 | 0.144 | -0.059 | 0.189 | -0.078 | -0.006 | **0.917** |

Larcker [101] technique (see Table 4), where the criterion for establishing discriminant validity involves comparing the average variance extracted (AVE) with the square of correlations or the square root of AVE with correlations [90]. The second method, depicted in Table 5, compares the square root of AVE with the correlation values. When the square root of AVE (shown on the diagonals) is higher than the values in the respective construct's columns and rows, we can conclude that the measures are discriminant. Further, discriminant validity is also examine using HTMT ratio. The HTMT ratio serves as the basis for establishing discriminant validity. Henseler et al. [102] and Kock [103] recommended a liberal threshold of 0.90 or less, whereas Kline [104] suggested a threshold of 0.85 or less. The conditions are satisfied by the data used for this study. As per Tables 4 and 5 the values on the diagonals exceed the values in their corresponding columns and rows, indicating satisfactory discriminant validity for the utilized metrics in this study.

**Test of hypothesis.** The results of hypothesis testing are presented in Table 6.

The analysis of the factors influencing technology perceptions has produced significant results (See Fig 6). Hypothesis 1 (H1) found that technology anxiety is positively linked to perceived usefulness (0.101), suggesting that more anxious individuals may value technology's benefits more. Hypothesis 2 (H2) showed that technology anxiety also increases perceived ease of use (0.188), possibly because anxious users work harder to master technology. Hypothesis 3 (H3) and Hypothesis 4 (H4) revealed that self-efficacy significantly enhances perceived usefulness (0.312) and ease of use (0.170), respectively, indicating that confidence in using technology leads to better perceptions of it. Hypothesis 5 (H5) indicated a slight but significant relationship between individual innovativeness and perceived usefulness (0.004). These

**Table 5. Discriminant validity (HTMT Ratio).**

|  | AU | BI | FC | INI | PEOU | PU | SE | SI | TA |
|---|---|---|---|---|---|---|---|---|---|
| AU |  |  |  |  |  |  |  |  |  |
| BI | 0.684 |  |  |  |  |  |  |  |  |
| FC | 0.687 | 0.689 |  |  |  |  |  |  |  |
| INI | 0.428 | 0.465 | 0.665 |  |  |  |  |  |  |
| PEOU | 0.568 | 0.773 | 0.708 | 0.701 |  |  |  |  |  |
| PU | 0.58 | 0.383 | 0.76 | 0.618 | 0.784 |  |  |  |  |
| SE | 0.41 | 0.384 | 0.726 | 0.542 | 0.471 | 0.536 |  |  |  |
| SI | 0.389 | 0.365 | 0.568 | 0.671 | 0.629 | 0.648 | 0.61 |  |  |
| TA | 0.223 | 0.154 | 0.232 | 0.236 | 0.355 | 0.232 | 0.19 | 0.13 |  |

**Table 6. Test of hypothesis.**

| Hypotheses | Estimate |
|---|---|
| H1: Technology Anxiety → Perceived Usefulness | .101*** (.033) |
| H2: Technology Anxiety → Perceived Ease of Use | .188*** (.041) |
| H3: Self-efficacy → Perceived Usefulness | .312*** (.060) |
| H4: Self-efficacy → Perceived Ease of Use | .170** (.073) |
| H5: Individual Innovativeness→ Perceived Usefulness | .004** (.068) |
| H6: Individual Innovativeness→ Perceived Ease of Use | .281*** (.088) |
| H7: Facilitating Conditions → Perceived Usefulness | -.199*** (.062) |
| H8: Facilitating Conditions → Perceived Ease of Use | -.107 (.079) |
| H9: Social Influence → Perceived Usefulness | .411*** (.097) |
| H10: Social Influence → Perceived Ease of Use | .261** (.121) |
| H11: Perceived Ease of Use → Perceived Usefulness | -.164*** (.053) |
| H12: Perceived Usefulness → Attitude towards using Agriculture 5.0 | .083* (.048) |
| H13: Perceived Ease of Use → Attitude Towards Using Agriculture 5.0 | .189*** (.037) |
| H14: Attitude towards using → Behavioral Intention to Use Agriculture 5.0 | .858*** (.138) |

Notes: Standard errors of coefficient estimates are in the parentheses.

***, **, and * denote 1%, 5% and 10% level of significance, respectively.

insights highlight the nuanced relationship between psychological factors and technology adoption, informing strategies to improve technology engagement.

Similarly, hypothesis 6 (H6) shows that individual innovativeness greatly improves perceived ease of use (0.281), suggesting that innovative people find technology more user-friendly. Hypothesis 7 (H7) unexpectedly indicates that facilitating conditions may decrease perceived usefulness (-0.199), a result that calls for further study. Hypothesis 8 (H8) suggests that facilitating conditions have no significant effect on ease of use (-0.107). Hypothesis 9 (H9) confirms a strong positive impact of social influence on perceived usefulness (0 .411), while Hypothesis 10 (H10) finds that social influence also increases perceived ease of use (0.261), albeit less so than usefulness. These insights reveal the complex interplay of personal and social factors in the adoption of technology.

Moreover, hypothesis 11 (H11) indicates a surprising negative link between ease of use and perceived usefulness (-0.164), hinting that simpler technologies might be undervalued. Hypothesis 12 (H12) finds that perceived usefulness has a slight positive impact on attitudes towards Ag 5.0 (0.083). Hypothesis 13 (H13) shows a significant positive relationship between ease of use and attitude (0.189), emphasizing the role of user-friendliness. Hypothesis 14 (H14) reveals a strong connection between positive attitudes and the intention to use Ag 5.0 (0.858), highlighting the importance of positive perceptions in adoption intentions. Tama et al. [105]

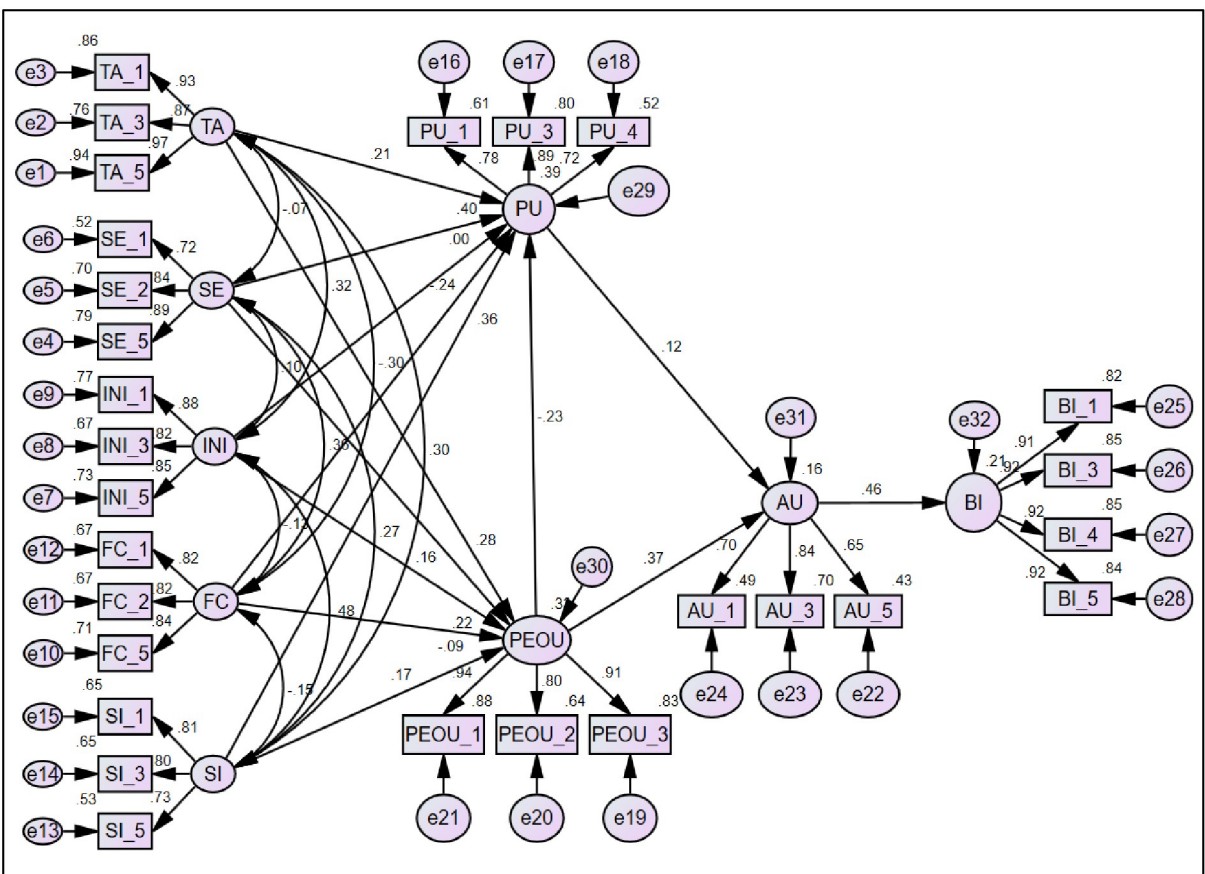

**Fig 6. SEM for direct, indirect and mediation analysis.**

have found the significant impact of the complexity and compatibility on farmers attitude for adopting conservation agriculture program in Bangladesh.

**Mediational analysis.** Mediation analysis explores the significant impact of mediating variables on the dependent or explanatory variables. In this analysis, the independent variable, referred to as X, is assumed to influence a mediator (M), which in turn affects a dependent variable (Y), based on the model structure [106]. To demonstrate the mediation relationship, the Sobel Test was used (see Table 7). The mediation analysis shows that direct and indirect effect on model. In this study six mediation analysis has investigated such as, TA→ PEOU→ PU, SE→ PEOU→ PU, INI→ PEOU→ PU, FC→ PEOU→ PU, SI→ PEOU→ PU, PEOU→ PU→ AU.

From the result we concluded that the perceived ease of use (PEOU) does not have a statistically significant mediating effect on the external variables technology (TA, SE, INI, FC, SI) and perceived usefulness (PU). Similarly, the perceived usefulness (PU) does not mediate the relationship between perceived ease of use (PEOU) and attitude towards use (AU). Farmers are not well trained to accept and use the technology in their farming system which shows that the fear of anxiety. The study by Pillai and Sivathanu [93] shows a similar result where anxiety is one of the barriers for technology adoption. Farmers feel that their skill and attitude to use the technology will make it easy to use the technology, the similar result was shown by Zarafshani et al. [107]. The social influence has greater impact in the perceived ease of use and perceived

**Table 7. Result of indirect effects on SOBEL test examining the mediating relationship.**

| Hypothesis | | | Mediating effect | | | |
|---|---|---|---|---|---|---|
| | | | b | Sb | $t_b$ | Sobel Test |
| TA→ PEOU→ PU | a | 0.275 | 0.088 | 0.049 | 1.786 | 1.7358 |
| | Sa | 0.038 | | | | |
| | $t_a$ | 6.784 | | | | |
| SE→ PEOU→ PU | a | 0.181 | 0.088 | 0.049 | 1.786 | 1.5161 |
| | Sa | 0.064 | | | | |
| | $t_a$ | 2.826 | | | | |
| INI→ PEOU→ PU | a | 0.452 | 0.088 | 0.049 | 1.786 | 1.7321 |
| | Sa | 0.069 | | | | |
| | ta | 6.592 | | | | |
| FC→ PEOU→ PU | a | -0.159 | 0.088 | 0.049 | 1.786 | -1.4399 |
| | Sa | 0.066 | | | | |
| | ta | -2.386 | | | | |
| SI→ PEOU→ PU | a | 0.472 | 0.088 | 0.049 | 1.786 | 1.7235 |
| | Sa | 0.077 | | | | |
| | ta | 6.138 | | | | |
| PEOU→ PU→ AU | a | 0.088 | 0.101 | 0.049 | 2.056 | 1.3541 |
| | Sa | 0.049 | | | | |
| | ta | 1.786 | | | | |

usefulness in the Italian farmers but it contradicts with the study [108]. The effect of perceived ease of use on perceived usefulness might depend on the area of application and therefore no statistical data was shown in the study similar in line with Michels et al. [109]. Similarly, Hua and Wang [110] found that the perceived usefulness have no impact on the consumers' purchasing intention for energy-efficient appliances. While Tama et al. [105] by using extend theory of planned behavior have found that increased level of knowledge can improve the farmers' intention to adopt conservation agriculture.

**Issues of nonlinear effects, endogeneity and unobserved heterogeneity.** According to authors like Hair et al. [111], Sarstedt et al. [112] and Vaithilingam et al. [113], researchers should consider potential nonlinear effects, endogeneity and unobserved heterogeneity in their structural models. Nonlinear effects in PLS-SEM involve examining relationships that are not strictly linear [114]. Researchers can introduce quadratic terms (squared predictor variables) or interaction terms to capture more complex relationships [115,116]. To test for nonlinearity, researchers can perform Ramsey's [117] RESET test on the latent variable scores in the path model's partial regressions [111]. Svensson et al. [118] and Memom et al. [119] recommend using bootstrapping techniques to map nonlinear effects in the model and test their statistical significance. In the model (see Table 8), the p-value between behavioral intention and continuance intention is higher than 0.05, indicating no linear relationship between these variables in the data set.

In a PLS-SEM analysis with an explanatory research perspective, it is crucial to test for endogeneity [111,120]. Endogeneity occurs when predictor variables are correlated with error terms, potentially leading to biased estimates [121]. This often happens when a construct that correlates with one or more predictor constructs and the dependent construct is omitted from the partial regression of the PLS path model [111]. The Gaussian copula approach is a systematic method to check for endogeneity issues [113,122]. The Gaussian copula (GC) technique

**Table 8. Fit Indices for linearity.**

|  | Original sample (O) | Sample mean (M) | Standard deviation (STDEV) | T statistics (\|O/STDEV\|) | P values |
|---|---|---|---|---|---|
| QE (AU) -> BI | -0.083 | -0.082 | 0.068 | 1.22 | 0.222 |
| QE (PEOU) -> AU | -0.035 | -0.029 | 0.091 | 0.392 | 0.695 |
| QE (PEOU) -> PU | 0.013 | 0.014 | 0.086 | 0.155 | 0.877 |
| QE (PU) -> AU | -0.014 | -0.013 | 0.066 | 0.207 | 0.836 |
| QE (SI) -> PEOU | 0.149 | 0.152 | 0.08 | 1.869 | 0.062 |
| QE (SI) -> PU | 0.134 | 0.148 | 0.079 | 1.69 | 0.091 |
| QE (FC) -> PEOU | 0.033 | 0.029 | 0.068 | 0.48 | 0.631 |
| QE (FC) -> PU | -0.084 | -0.093 | 0.11 | 0.757 | 0.449 |
| QE (INI) -> PEOU | -0.129 | -0.123 | 0.07 | 1.855 | 0.064 |
| QE (INI) -> PU | -0.032 | -0.01 | 0.07 | 0.454 | 0.650 |
| QE (SE) -> PEOU | 0.009 | 0.001 | 0.052 | 0.176 | 0.860 |
| QE (SE) -> PU | 0.04 | 0.024 | 0.061 | 0.667 | 0.505 |
| QE (TA) -> PEOU | 0.494 | 0.476 | 0.116 | 4.247 | 0.000 |
| QE (TA) -> PU | 0.115 | 0.085 | 0.135 | 0.85 | 0.395 |

requires a p-value greater than 0.05 to indicate no endogeneity [123,124]. In our analysis, we ran a Gaussian Copula test and found that, except for the relationship between GC (TA) and PEOU, the other 13 hypotheses showed no endogeneity issues, as their p-values were insignificant. Due to the complexity of the model, it is challenging to show multiple Gaussian Copulas (two, three, four, etc.). Thus, we present and report the final model of the Gaussian copula run (see Fig 7). Apart from GC (TA) ≥ PEOU, all other variables show insignificant p-values, indicating no endogeneity issues. Hair et al. [111] also noted that endogeneity assessment is relevant only when the research focus is explanatory, rather than on PLS-SEM's causal-predictive nature. Since this study focuses on causal predictive characteristics, a detailed endogeneity assessment is not conducted and is left for future research.

Unobserved heterogeneity occurs when data subgroups produce significantly different model estimates, potentially leading to misleading results if the model is estimated based on the entire dataset [111,125]. Therefore, PLS-SEM analyses should check for unobserved heterogeneity to determine if analyzing the entire dataset is appropriate [126]. Using information criteria from a finite mixture PLS [112,127], researchers can identify the number of segments to be extracted [111,126,128]. Becker et al. [125] recommend running the PLS prediction-oriented segmentation procedure to reveal segment structures when heterogeneity is at a critical level. Additionally, Ringle et al. [129] suggest identifying suitable explanatory variables that characterize the identified segments. We checked for unobserved heterogeneity in our dataset using various criteria (see Table 9). Our AIC, AIC3, AIC4, BIC, CAIC, and HQ criteria indicate that a third segment is appropriate, showing heterogeneity at a critical level. In this context, a PLS-SEM moderator [130,131] or multigroup analysis [128,132], combined with a measurement invariance assessment [133], is suggested when appropriate. As suggested by Hair et al. [111] and Vaithilingam et al. [113] detailed analysis can further identify specific findings, conclusions, and implications.

## 4. Conclusion and recommendations

This research investigates the farmers' behavioral intention for Agriculture 5.0. in Nepal. Specially, by using the SEM method, the study identifies the factors that influence the adoption of Agriculture 5.0 and proposes managerial solutions to promote Ag 5.0. The Results reveal that

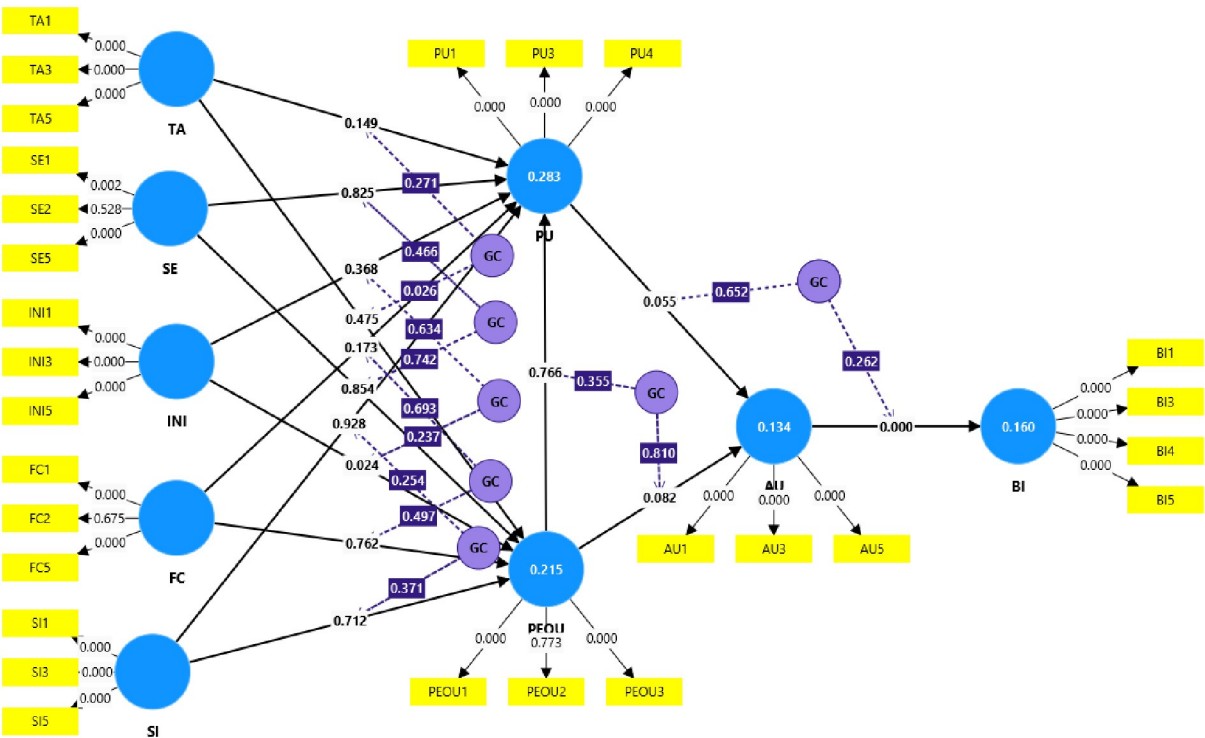

**Fig 7. Final model of the Gaussian copula.**

technological anxiety, self-efficacy, and social influence significantly impact the perceived usefulness and perceived ease of use of Ag 5.0-related technologies. This suggests on-farm training and educational programs that build confidence in farmers to adopt digital technologies should be provided on a routine basis by governmental agencies.

**Table 9. Fit indices for a one- to three-segment solution.**

|  | Number of Segments | | |
|---|---|---|---|
| Criteria | 1 | 2 | 3 |
| AIC | 2886.6 | 2651.72 | 2408.992 |
| AIC$_3$ | 2904.6 | 2688.72 | 2464.992 |
| AIC$_4$ | 2922.6 | 2725.72 | 2520.992 |
| BIC | 2951.438 | 2784.998 | 2610.711 |
| CAIC | 2969.438 | 2821.998 | 2666.711 |
| HQ | 2912.634 | 2705.233 | 2489.985 |
| MDL$_5$ | 3354.791 | 3614.112 | 3865.586 |
| LnL | -1425.3 | -1288.86 | -1148.5 |
| EN | na | 0.972 | 0.685 |
| NFI | na | 0.98 | 0.658 |
| NEC | na | 7.491 | 85.347 |

*Note*: AIC: Akaike's information criterion; AIC$_3$: modified AIC with factor 3; AIC$_4$: modified AIC with factor 4; BIC: Bayesian information criteria; CAIC: consistent AIC; HQ: Hannan Quinn criterion; MDL$_5$: minimum description length with factor 5; LnL: Log Likelihood; EN: entropy statistic; NFI: non-fuzzy index; NEC: normalized entropy criterion; na: not available; numbers in bold indicate the best outcome per segment retention criterion.

Furthermore, the study shows that the attitude toward Ag 5.0 significantly influences the behavioral intention to use it. Therefore, attitude changing strategies like method and result demonstration of using new technology, high-technology farm visit for the beginners, and subsides in high-technology driven equipment and machinery should be provided by the public sector to inculpate the positive attitudes towards Ag 5.0. In addition, farmers have optimistic views regarding the alternatives to overcome the perceived challenges. Developing farmer-based program to build confidence on dealing with risk of adopting new technology, extensive R& D to develop efficient technology fulfilling the local demand, subsides, and easy credit access are crucial in promoting extensive adoption of Ag 5.0. Promoting modernization and mechanization through technology-based farming is a pave to achieve the milestone stipulated in Agriculture Development Strategy and National Agricultural Development Policy of Nepal.

The findings offer insights to multiple stakeholders, such as the Ministry of Agriculture and Livestock Development, National Planning Commission, government bodies at provincial and local levels, decision-makers, development partners, and farmers' organizations. These insights include developing and implementing awareness raising program, developing digital infrastructure, offering necessary training and follow-up support for technology adoption, and establishing monitoring and coordination mechanisms to facilitate technology diffusion among most farmers.

There are some limitations to this study. First, the research was conducted in a single district of Nepal. Therefore, it may not represent the national-scale analysis. We recommend readers to be caution when generalizing results to the diverse geography of Nepal. Second, since the analysis is cross-sectional there can be potential omitted variable bias and expanding construct can mitigate bias. Future research can be conducted to address these limitations of the present study. Moreover, there is scope in considering additional factors, like technology exposure frequency, inclusion of managerial staff in the sample to better understand impact on perceived usefulness and ease of use in future research. Similarly, conducting a welfare analysis of governmental policies promoting Ag 5.0 adoption can also be an important area for future research.

## Author Contributions

**Conceptualization:** Nitesh Mishra, Tek Maraseni, Niranjan Devkota, Ghanashyam Khanal, Udaya Raj Paudel.

**Data curation:** Nitesh Mishra, Ghanashyam Khanal, Devid Kumar Basyal, Ranjana Kumari Danuwar.

**Formal analysis:** Nitesh Mishra, Nabin Bhandari, Niranjan Devkota, Ghanashyam Khanal, Devid Kumar Basyal, Ranjana Kumari Danuwar.

**Funding acquisition:** Nitesh Mishra, Udaya Raj Paudel.

**Investigation:** Nitesh Mishra, Niranjan Devkota, Udaya Raj Paudel.

**Methodology:** Nitesh Mishra, Tek Maraseni, Niranjan Devkota, Ghanashyam Khanal, Devid Kumar Basyal, Udaya Raj Paudel.

**Project administration:** Niranjan Devkota, Ghanashyam Khanal.

**Resources:** Nabin Bhandari, Tek Maraseni, Biswash Bhusal.

**Software:** Nitesh Mishra, Nabin Bhandari, Devid Kumar Basyal, Ranjana Kumari Danuwar.

**Supervision:** Niranjan Devkota, Devid Kumar Basyal.

**Validation:** Nabin Bhandari, Tek Maraseni, Biswash Bhusal.

**Visualization:** Nabin Bhandari, Biswash Bhusal.

**Writing – original draft:** Nitesh Mishra, Nabin Bhandari, Niranjan Devkota, Biswash Bhusal, Devid Kumar Basyal, Udaya Raj Paudel, Ranjana Kumari Danuwar.

**Writing – review & editing:** Tek Maraseni, Niranjan Devkota, Ghanashyam Khanal, Biswash Bhusal, Devid Kumar Basyal, Udaya Raj Paudel, Ranjana Kumari Danuwar.

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
