## [Decision Letter · Decision Letter 0]

17 Jan 2024

PONE-D-23-25144Technology in Farming: Unleashing Farmers’ Behavioral Intention for the Adoption of Agriculture 5.0PLOS ONE

Dear Dr. Devkota,

Thank you for submitting your manuscript to PLOS ONE. After careful consideration, we feel that it has merit but does not fully meet PLOS ONE’s publication criteria as it currently stands. Therefore, we invite you to submit a revised version of the manuscript that addresses the points raised during the review process.

We look forward to receiving your revised manuscript.

Kind regards,

Md. Monirul Islam, PhD

Academic Editor

PLOS ONE

Journal Requirements:

6. We note that your Data Availability Statement is currently as follows: “All relevant data are within the manuscript and its Supporting Information files.”

If there are ethical or legal restrictions on sharing a de-identified data set, please explain them in detail (e.g., data contain potentially sensitive information, data are owned by a third-party organization, etc.) and who has imposed them (e.g., an ethics committee). Please also provide contact information for a data access committee, ethics committee, or other institutional body to which data requests may be sent. If data are owned by a third party, please indicate how others may request data access."

7. We note that Figure 3 in your submission contain map images which may be copyrighted. All PLOS content is published under the Creative Commons Attribution License (CC BY 4.0), which means that the manuscript, images, and Supporting Information files will be freely available online, and any third party is permitted to access, download, copy, distribute, and use these materials in any way, even commercially, with proper attribution. For these reasons, we cannot publish previously copyrighted maps or satellite images created using proprietary data, such as Google software (Google Maps, Street View, and Earth). For more information, see our copyright guidelines: http://journals.plos.org/plosone/s/licenses-and-copyright.

    a. You may seek permission from the original copyright holder of Figure 3 to publish the content specifically under the CC BY 4.0 license. 

Reviewers' comments:

Reviewer's Responses to Questions

**Comments to the Author**

1. Is the manuscript technically sound, and do the data support the conclusions?

Reviewer #1: No

Reviewer #2: No

Reviewer #3: Yes

2. Has the statistical analysis been performed appropriately and rigorously? 

Reviewer #1: Yes

Reviewer #2: No

Reviewer #3: Yes

3. Have the authors made all data underlying the findings in their manuscript fully available?

Reviewer #1: Yes

Reviewer #2: Yes

Reviewer #3: No

4. Is the manuscript presented in an intelligible fashion and written in standard English?

Reviewer #1: Yes

Reviewer #2: No

Reviewer #3: Yes

5. Review Comments to the Author

Reviewer-1

Hi,

P.10: If Survey is used, why the sampling technique is purposive? It should be random. Please check and provide justifiable reasons.

p.11: why 14 respondents? please explain clearly in the research method section.

p.13: Figure 4. n=???. add the amount of n.

p.18: The references of "Zarafshani et al., 2020, did not exist in the final reference list of the paper.

p.19: Conclusion is weak and need to be strengthened.

p.19: Please add some applicable suggestions based on the research findings.

p. 20-25: References list should be check with the guidelines of PLOS ONE. some references did not report correctly. some information was missed.

There are some mistakes in in-text citation of the manuscript as well that it need to be double checked and revised due to the PLOS ONE guidelines

Reviewer-2

The study titled Technology in Farming: Unleashing Farmers’ Behavioral Intention for the Adoption of Agriculture 5.0' aimed to explore the factors influencing farmer’s behavioral intension for Agriculture 5.0, identifies implementation obstacles and provides managerial solutions to promote Ag 5.0 in Madhesh Province, Nepal, using the Technology Acceptance Model (TAM) and

Structural Equation Model (SEM). The findings reveal that farmers perceive training programs, government assistance, and subsidies are helpful in overcoming challenges associated with adopting Ag 5.0. In my opinion, the study lacks in certain major ways that cast doubt on the validity of the results. My criticism centres mostly around the statistical result of the research study, the over generalisation of the findings, and unfounded claims (as discussion section is missing !).

A result statistically presented as relevant with a p-significance greater than 0.050 (e.g., “Individual innovativeness also affects the perceived usefulness (β =0.004, p>0.05) and perceived ease of use (β =0.281, p>0.01)” is definitely a major flaw that leads to misinterpretation of the final results. The authors definitely lack basic knowledge in interpreting statistical data, which is a pity because the chosen topic is of interest. In my opinion this study cannot be published in any respectable journal, especially PlosOne.

Reviewer-3

The manuscript deals with an intriguing and valuable subject matter. Overall, the article is well-written and well-structured. The research investigates farmers' behavioral intentions towards adopting Agriculture 5.0, identifies influencing factors, explores implementation obstacles, and proposes some management-level solutions to promote Ag 5.0. However, the article will benefit from the following suggestions:

1. I feel that the empirical analysis is weak. In particular, the author does not conduct sufficient robustness checks to test the sensitivity of the key findings. For SEM analysis, data must be normally distributed, which is not the case in the article.

2. The author recognizes that the adoption of latest farming technologies is low in Nepal. However, this needs to be elaborated with data and statistics to justify this research. In addition, only mentioning a lack of literature does not fully justify this research. The author needs to further highlight the novelty and contribution of this work aligning with policies and programs concerning Ag 5.0 and food security. A separate literature review section can be tried.

3. The manuscript largely deals with farmers’ attitude for adopting Ag 5.0 related technologies but does not draw significantly from the current literature. For this purpose, the author can draw some significant findings (in support of in contrast) with the popular assumptions. The articles listed below can be a good starting point.

4. Dataset was collected from 258 farmers. However, the research methodology did not justify the use of quantitative methods within a particular epistemological stream (i.e., positivism and quantitative). Why farmers’ perceptions and opinions/beliefs are theoretically best suited to address the research hypotheses needs to be mentioned. Why were managerial staffs were not included in the study?

5. The article is focused more on results but less on discussion. Discussion and implications of the results should be elaborated, especially in relation to Ag 5.0. It will also help to make the article relevant to the wider international readership of the journal.

6. The conclusion section is well-written and summarizes things up. However, the first paragraph says farmers have a fear of using new technology while the second paragraph says farmers are ready to adopt new technologies of Ag 5.0. This should be justified why these are not contradictory.

7. The manuscript does contain some typos and language errors. A careful revision, especially in the tables and figures, will improve its readability.

8. Please consider the literature below in your revision:

1. Tama, R. A. Z., Hoque, M. M., Liu, Y., Alam, M. J., & Yu, M. (2023). An Application of Partial Least Squares Structural Equation Modeling (PLS-SEM) to Examining Farmers’ Behavioral Attitude and Intention towards Conservation Agriculture in Bangladesh. Agriculture, 13(2), 503.

2. Hua, L.; Wang, S. Antecedents of Consumers’ Intention to Purchase Energy-Efficient Appliances: An Empirical Study Based on the Technology Acceptance Model and Theory of Planned Behavior. Sustainability 2019, 11, 2994

3. Tama, R.A.Z.; Ying, L.; Yu, M.; Hoque, M.; Adnan, K.M.; Sarker, S.A. Assessing farmers’ intention towards conservation agriculture by using the Extended Theory of Planned Behavior. J. Environ. Manag. 2021, 280, 111654

4. Alambaigi, A.; Ahangari, I. Technology Acceptance Model (TAM) As a Predictor Model for Explaining Agricultural Experts Behavior in Acceptance of ICT. Int. J. Agric. Manag. Dev. 2016, 6, 235–247.

5. Amin, M.; Rezaei, S.; Abolghasemi, M. User satisfaction with mobile websites: The impact of perceived usefulness (PU), perceived ease of use (PEOU) and trust. Nankai Bus. Rev. Int. 2014, 5, 258–274.n

6. PLOS authors have the option to publish the peer review history of their article (what does this mean?). If published, this will include your full peer review and any attached files.

Reviewer #1: No

Reviewer #2: No

Reviewer #3: No

---

## [Author Response · Author response to Decision Letter 0]

17 Mar 2024

Dear Reviewers and Editor,

We extend our gratitude to the reviewers and editor for providing constructive comments, which we greatly value. The opportunity to further enhance our paper is always appreciated, and we sincerely thank you for your insightful suggestions. We have diligently addressed your comments to the best of our ability in this revision, resulting in tangible improvements to the manuscript. We are optimistic that you will find both our responses and revisions to be satisfactory.

Reviewer 1

Comment: 1. P.10: If a Survey is used, why the sampling technique is purposive? It should be random. Please check and provide justifiable reasons.

Response: Thank you for your suggestion. Yes, you are right; we did conduct random sampling. Here's what we mean: the study district was purposively selected after discussions with staff from agricultural departments and experts. There are several reasons for selecting this district (agricultural hub in Nepal and many pilot projects are ongoing, etc.). After selecting this district, we delved deeper and found that there are 67,058 households engaged in the agricultural sector, which represents our population. Then, after applying the formula (mention the name here), the required sample size was estimated to be 246. Adding 5% for non-respondents (12) as an error margin, our total sample size taken for the study is 258. These sample households were randomly selected using a random table.

Comment: 2. p.11: why 14 respondents? Please explain clearly in the research method section. 

Response: We appreciate your time spent reviewing our manuscript. Pretesting has been conducted prior to commencing the actual survey with the respondents. The supervisor's suggestion to select 5% of the total sample as criteria for pretesting the questionnaire was adhered to, ensuring satisfaction with the given number. The aim of pretesting is to assess the clarity and feasibility of the questionnaire, as well as to identify any necessary refinements. Following the completion of pretesting with 14 respondents, no further comments were observed. Based on this, both the department and the supervisor have approved proceeding with the final survey. Thus, only 14 respondents were involved in the pretesting of the questionnaire.

Comment: 3. p.13: Figure 4. n=???. add the amount of n.

Response: Thank you for your constructive suggestion. We have added the total number of respondents equal to 256 in Figure 4. We apologize for this error. 

Comment: 4. p.18: The references of "Zarafshani et al., 2020, did not exist in the final reference list of the paper.

Response: We agree with you and in our revised manuscript we have added the reference list.

Zarafshani, K., Solaymani, A., Itri, M., Helms, M., and Sanjabi, S. 2020. Evaluating technology acceptance in agricultural education in Iran: A study of vocational agriculture teachers. Social sciences & humanities open. 2(2020) 100041. Available at https://doi.org/10.1016/j.ssaho.2020.100041. 

Comment: 5. P19. The conclusion is weak and needs to be strengthened.

Response: We agree to your comment and have revised the conclusion part in our new manuscript as follows: 

“This research investigates the farmers' behavioral intention for Agriculture 5.0 in Nepal. Specially, by using the SEM method, the study identifies the factors that influence the adoption of Agriculture 5.0 and proposes managerial solutions to promote Ag 5.0. The results reveal that technological anxiety, self-efficacy, and social influence significantly impact the perceived usefulness and perceived ease of use of Ag 5.0-related technologies. This suggests on-farm training and educational programs that build confidence in farmers to adopt digital technologies should be provided on a routine basis by governmental agencies. 

Furthermore, the study shows that the attitude toward Ag 5.0 significantly influences the behavioral intention to use it. Therefore, attitude changing strategies like method and result demonstration of using new technology, high-technology farm visit for the beginners, and subsides in high-technology driven equipment and machinery should be provided by the public sector to inculpate the positive attitudes towards Ag 5.0. In addition, farmers have optimistic views regarding the alternatives to overcome the perceived challenges. Developing farmer-based program to build confidence on dealing with risk of adopting new technology, extensive R& D to develop efficient technology fulfilling the local demand, subsides, and easy credit access are crucial in promoting extensive adoption of Ag 5.0. Promoting modernization and mechanization through technology-based farming is a pave to achieve the milestone stipulated in Agriculture Development Strategy and National Agricultural Development Policy of Nepal. 

The findings offer insights to multiple stakeholders, such as the Ministry of Agriculture and Livestock Development, National Planning Commission, government bodies at provincial and local levels, decision-makers, development partners, and farmers' organizations. These insights include developing and implementing awareness raising program, developing digital infrastructure, offering necessary training and follow-up support for technology adoption, and establishing monitoring and coordination mechanisms to facilitate technology diffusion among most farmers.

There are some limitations to this study. First, the research was conducted in a single district of Nepal. Therefore, it may not represent the national-scale analysis. We recommend readers to be caution when generalizing results to the diverse geography of Nepal. Second, since the analysis is cross-sectional there can be potential omitted variable bias and expanding construct can mitigate bias. Future research can be conducted to address these limitations of the present study. Moreover, there is scope in considering additional factors, like technology exposure frequency, inclusion of managerial staff in the sample to better understand impact on perceived usefulness and ease of use in future research. Similarly, conducting a welfare analysis of governmental policies promoting Ag 5.0 adoption can also be an important area for future research.”

Comment: 6. p.19: Please add some applicable suggestions based on the research findings.

Response: Thank you for your feedback. Applicable suggestions have been added in the conclusion and suggestion part. To be specific, as noted above, we have rewritten the conclusion part to include appropriate suggestion based on research findings. 

Comments: 7. p. 20-25: References list should be checked with the guidelines of PLOS ONE. some references did not report correctly. some information was missed.

There are some mistakes in in-text citation of the manuscript as well that it needs to be double checked and revised due to the PLOS ONE guidelines.

Response: Thank you for pointing out our errors. We have double-checked the references and edited them whenever necessary. 

Reviewer 2

 Comment:1. The study titled Technology in Farming: Unleashing Farmers’ Behavioral Intention for the Adoption of Agriculture 5.0' aimed to explore the factors influencing farmer’s behavioral intension for Agriculture 5.0, identifies implementation obstacles and provides managerial solutions to promote Ag 5.0 in Madhesh Province, Nepal, using the Technology Acceptance Model (TAM) andStructural Equation Model (SEM). The findings reveal that farmers perceive training programs, government assistance, and subsidies are helpful in overcoming challenges associated with adopting Ag 5.0. In my opinion, the study lacks in certain major ways that cast doubt on the validity of the results. My criticism centres mostly around the statistical result of the research study, the over generalization of the findings, and unfounded claims (as discussion section is missing !).

A result statistically presented as relevant with a p-significance greater than 0.050 (e.g., “Individual innovativeness also affects the perceived usefulness (β =0.004, p>0.05) and perceived ease of use (β =0.281, p>0.01)” is definitely a major flaw that leads to misinterpretation of the final results. The authors definitely lack basic knowledge in interpreting statistical data, which is a pity because the chosen topic is of interest. In my opinion this study cannot be published in any respectable journal, especially PlosOne.

Response: We are extremely sorry for this. The statistics are well presented in the result section, but it seems we had a typo error in abstract. We highly appreciate and regard this comment as feedback for future work. We have edited the abstract in the revised manuscript. Furthermore, we have included the relevance of finding in conclusion section. 

Reviewer 3

Comment: 1. I feel that the empirical analysis is weak. In particular, the author does not conduct sufficient robustness checks to test the sensitivity of the key findings. For SEM analysis, data must be normally distributed, which is not the case in the article.

Response: Thank you for highlighting the importance of the robustness check. Following feedback, we have meticulously reviewed the entire robustness check, addressing any previously overlooked elements and incorporating missing values, including a comprehensive normality test – both Kurtosis and Skewness value in result section. The normality test of the dataset use for the analysis reveals that the kurtosis value ranges from -1.473 to +3.472 (i.e. between -4 to +4) and its skewness value ranges from -1.443 to -0.196 (between -2 to +2). These findings suggest that the dataset used in the analysis does not exhibit any normality issues. This aligns with the criteria proposed by Hair et al. (2010), Bryne (2010) and Brown (2015) who assert that data is considered normal when kurtosis falls within the range of -7 to +7 and skewness is between -2 to +2 (see page 15). We have also shared additional insights to reaffirm the discriminant validity, we have introduced the HTMT Ratio, aligning with the criteria set by PLS-SEM. The HTMT ratio serves as the basis for establishing discriminant validity. Henseler et al. (2015) and Kock (2020) recommended a liberal threshold of 0.90 or less, whereas Kline (2015) suggested a threshold of 0.85 or less. The conditions are satisfied by the data used for this study. As a result of these enhancements, we are confident that the result section is now free from any further issues. 

Comment: 2. The author recognizes that the adoption of latest farming technologies is low in Nepal. However, this needs to be elaborated with data and statistics to justify this research. In addition, only mentioning a lack of literature does not fully justify this research. The author needs to further highlight the novelty and contribution of this work aligning with policies and programs concerning Ag 5.0 and food security. A separate literature review section can be tried.

Response: Thank you for your constructive feedback. Based on your suggestion and feedback we have incorporated the data and statistics to justify our research and have highlighted the contribution of this work. We focused on how the study align with policies and programs concerning agriculture 5.0 and food security in the context of Nepal. This study also provides valuable insights for policymakers, development partners, and farmers' organizations, enabling them to understand the factors influencing the readiness for Ag 5.0 adoption in Nepal and provide them alternative solution to overcome the hurdles associated with the implementation of Ag. 5.0 in Nepal. These findings and solutions could be applicable in many other developing countries with similar seriocomic settings. Based on other authors’ comment we have included the literature review within the text whenever necessary. If you still believe separate literature section would better polish this article we are happy to accomplish that. 

Comment: 3. The manuscript largely deals with farmers’ attitude for adopting Ag 5.0 related technologies but does not draw significantly from the current literature. For this purpose, the author can draw some significant findings (in support of in contrast) with the popular assumptions. The articles listed below can be a good starting point.

Response: We have reviewed the suggested literature and incorporated the pertinent ideas in our revised manuscript. We added the following new articles among others:

Tama, R. A. Z., Hoque, M. M., Liu, Y., Alam, M. J., & Yu, M. (2023). An Application of Partial Least Squares Structural Equation Modeling (PLS-SEM) to Examining Farmers’ Behavioral Attitude and Intention towards Conservation Agriculture in Bangladesh. Agriculture, 13(2), 503.

Hua, L.; Wang, S. Antecedents of Consumers’ Intention to Purchase Energy-Efficient Appliances: An Empirical Study Based on the Technology Acceptance Model and Theory of Planned Behavior. Sustainability 2019, 11, 2994

Tama, R.A.Z.; Ying, L.; Yu, M.; Hoque, M.; Adnan, K.M.; Sarker, S.A. Assessing farmers’ intention towards conservation agriculture by using the Extended Theory of Planned Behavior. J. Environ. Manag. 2021, 280, 111654

Alambaigi, A.; Ahangari, I. Technology Acceptance Model (TAM) As a Predictor Model for Explaining Agricultural Experts Behavior in Acceptance of ICT. Int. J. Agric. Manag. Dev. 2016, 6, 235–247.

Amin, M.; Rezaei, S.; Abolghasemi, M. User satisfaction with mobile websites: The impact of perceived usefulness (PU), perceived ease of use (PEOU) and trust. Nankai Bus. Rev. Int. 2014, 5, 258–274.n

Comment: 4. Dataset was collected from 258 farmers. However, the research methodology did not justify the use of quantitative methods within a particular epistemological stream (i.e., positivism and quantitative). Why farmers’ perceptions and opinions/beliefs are theoretically best suited to address the research hypotheses needs to be mentioned. Why were managerial staffs were not included in the study?

Response: Thank you for the comments. We have added the importance of studying perceptions and beliefs to identify the farmers’ behavioral intention for the adoption of agriculture 5.0 in the context of Nepal. Furthermore, we were assigned to include farmers as the subject of study considering farmers' perceptions or opinions when introducing new agricultural technology is essential for maximizing adoption rates, ensuring technology relevance and success, improving productivity, promoting sustainability, and addressing socio-economic factors that influence technology uptake. Research including managerial staff can be another potential research question for future research. Adding on it, whatever policy we develop, without farmers' willingness, they cannot be adopted, as farmers are the implementers of technology etc. are the issue for agenda for upcoming research as this paper cannot capture all these aspects. Thus, we have pointed out the importance of studying farmers beliefs and perceptions in new manuscript. 

Comment: 5. The article is focused more on results but less on discussion. Discussion and implications of the results should be elaborated, especially in relation to Ag 5.0. It will also help to make the article relevant to the wider international readership of the journal.

Response: We agree to your comment and believe that enriching the discussion part will make the article more relevant to wider international readers of this reputed journal. Whenever necessary and relevant we have elaborated our discussion in the revised manuscript. We added the following in discussion section: 

“The analysis of the factors influencing technology perceptions has produced significant results. Hypothesis 1 (H1) found that technology anxiety is positively linked to perceived usefulness (0.101), suggesting that more anxious individuals may value technology's benefits more. Hypothesis 2 (H2) showed that technology anxiety also increases perceived ease of use (0.188), possibly because anxious users work harder to master technology. Hypothesis 3 (H3) and Hypothesis 4 (H4) revealed that self-efficacy significantly enhances perceived usefulness (0.312) and ease of use (0.170), respectively, indicating that confidence in using technology leads to better perceptions of it. Hy

---

## [Decision Letter · Decision Letter 1]

10 Jul 2024

PONE-D-23-25144R1Technology in Farming: Unleashing Farmers’ Behavioral Intention for the Adoption of Agriculture 5.0PLOS ONE

Dear Dr. Devkota,

Thank you for submitting your manuscript to PLOS ONE. After careful consideration, we feel that it has merit but does not fully meet PLOS ONE’s publication criteria as it currently stands. Therefore, we invite you to submit a revised version of the manuscript that addresses the points raised during the review process.

We look forward to receiving your revised manuscript.

Kind regards,

Md. Monirul Islam, PhD

Academic Editor

PLOS ONE

Journal Requirements:

Additional Editor Comments:

Dear Author

Thank you so much for your effort. The manuscript can be accepted after robustness check of your model. Thank you.

Reviewers' comments:

Reviewer's Responses to Questions

**Comments to the Author**

1. If the authors have adequately addressed your comments raised in a previous round of review and you feel that this manuscript is now acceptable for publication, you may indicate that here to bypass the “Comments to the Author” section, enter your conflict of interest statement in the “Confidential to Editor” section, and submit your "Accept" recommendation.

Reviewer #1: All comments have been addressed

Reviewer #3: All comments have been addressed

Reviewer #4: (No Response)

Reviewer #5: All comments have been addressed

2. Is the manuscript technically sound, and do the data support the conclusions?

Reviewer #1: Yes

Reviewer #3: Partly

Reviewer #4: Partly

Reviewer #5: Yes

3. Has the statistical analysis been performed appropriately and rigorously? 

Reviewer #1: Yes

Reviewer #3: No

Reviewer #4: No

Reviewer #5: Yes

4. Have the authors made all data underlying the findings in their manuscript fully available?

Reviewer #1: Yes

Reviewer #3: No

Reviewer #4: Yes

Reviewer #5: Yes

5. Is the manuscript presented in an intelligible fashion and written in standard English?

Reviewer #1: Yes

Reviewer #3: Yes

Reviewer #4: Yes

Reviewer #5: Yes

6. Review Comments to the Author

Reviewer #1: Hi,

Pleas double check the reference list that all of the references were cited correctly.

For example I saw that one of the references was not cited completely that is needed to be corrected as below:

Salehi, S., Rezaei-Moghaddam, K., & Ajili, A. (2012). Extension of grid soil sampling technology: application of extended Technology Acceptance Model (TAM). Journal of Research in Agriculture. 1, 078-087.

Reviewer #3: Thank you so much for submitting the revised version of the manuscript. Previous comments have been mostly addressed. However, although the data normality test has been performed, the data robustness check still lacks sufficiency.

In the PLS-SEM approach, robustness check can be performed in three ways:

1. the non-linear effects

2. Unobserved heterogeneity

3. Endogeneity

Reviewer #4: Thank you very much for giving me the opportunit to review the paper!

I am sorry to say that ins current form the paper cannot be accepted for publication. In fact, I recommend the paper to be rejected. First of all, the paper does not provide a clear research gap or its contribution to the literature. Second, the framework is random and falsely used. In consequence, the implications given by the results are not valid. Hence, the addition of the paper to the literature is questionable.

Major points:

The introduction is not clearly written and understandable. The authors jump back and forth between various themes. For instance, they start explaining on page 2 the concept of Ag 5.0 and then go back to the Green Evolution and then again Ag 5.0. Hence, there is no clear structure in the introduction and therefore the research gap is unclear. I was not able to detect based on the literatur provided what is the research gap and what is the contribution of this paper. This needs to be backed up by literature. Furthermore, for the authors it is not of interest, what the authors belief (p. 3) regarding farmers' beliefs. This needed to be backed up by literature. Hence, the complete introduction needs to be rewritten.

The framework is very unclear. Why does the authors try to combine elements from the UTAUT (Facilitating conditions, Social Influence) and the TAM (Perceived Usefulness)? The UTAUT is already as stated by its name (Unified Theory of Acceptance and Use of Technology) the essence of 8 distinct theories, including the TAM. Hence. the framework feels random, even though back up by another reference. It is very questionable why the authors have not chosen to follow the path of the UTAUT and extend it with technology anxiety. To highlight this puzzling choice by the authors: According to the UTAUT, it is expected that Social Influence has an an effect on the Intention to use Technology. Here, it influences Perceived Ease of Use and Perceived Usefulness.

The part on Structural equation modelling misses most important parts regarding interal and external validity of the model. Furthermore information is missing on the used statistical software.

Regarding the results, looking at the Cronbachs Alpha and Composite reliabiltiy. The values for Behavioral Intention exceed 0.95 which usually points semantically to close statements. Hence, these statements do not measure a common construct but just represent the same sentence all over again. For discriminant validity, it is always recommended to provide the CI95 value, which is missing. Effect sizes f2 are missing. Furthermore, the authors rely only statistical significance. Most effect sizes of the path coefficient can be considered to be very low. Hence, the practical relevance is missing and not discussed by the authors sufficiently. Lastly, as usual one should test for quadrat effects and endogeneity via Gaussian copulas in SEM. Both is missing.

Minor points

Term significant should only be used in connection with statistical significane and inference.

Figure 4: Barriers are not fully written out (Lack of Adquate Infrastructure &..).

Reviewer #5: (No Response)

7. PLOS authors have the option to publish the peer review history of their article (what does this mean?). If published, this will include your full peer review and any attached files.

Reviewer #1: **Yes: **Kurosh Rezaei-Moghaddam

Reviewer #3: **Yes: **Riffat Ara Zannat Tama, PhD

Reviewer #4: No

Reviewer #5: **Yes: **Nalini Arumugam

---

## [Author Response · Author response to Decision Letter 1]

27 Jul 2024

Dear Editor,

We are delighted to resubmit, 2nd response to the reviewer, our manuscript titled “Technology in Farming: Unleashing Farmers’ Behavioral Intention for the Adoption of Agriculture 5.0” to your esteemed journal.

Having carefully reviewed and incorporated all feedback, we believe the revisions have significantly strengthened the paper. Each comment has been addressed with thoughtful logic, enhancing the overall robustness of the manuscript. I have attached entire required document as mentioned in the comments and feedback section of decision letter. 

We are grateful for the opportunity to improve our work and eagerly anticipate any further feedback you may provide. Thank you for your time and consideration.

Sincerely yours’

Niranjan Devkota, PhD 

Head, Resaerch Management Cell, Kathmandu Model College, Nepal

Associate Researcher, National Planning Commission, Government of Nepal

---

## [Editor Report · Decision Letter 2]

1 Aug 2024

Technology in Farming: Unleashing Farmers’ Behavioral Intention for the Adoption of Agriculture 5.0

PONE-D-23-25144R2

Dear Dr. Niranjan Devkota,

We’re pleased to inform you that your manuscript has been judged scientifically suitable for publication and will be formally accepted for publication once it meets all outstanding technical requirements.

Kind regards,

Md. Monirul Islam, PhD

Academic Editor

PLOS ONE

Additional Editor Comments (optional):

Dear Author

We greatly appreciate the effort and dedication you have demonstrated in addressing the reviewers' comments and making the necessary revisions to enhance the quality of your work. Your thorough and thoughtful responses have significantly improved the clarity, depth, and overall contribution of your manuscript to the field. Congratulations on this accomplishment.
---

## [Editor Report · Acceptance letter]

12 Aug 2024

PONE-D-23-25144R2 

PLOS ONE

Dear Dr. Devkota, 

I'm pleased to inform you that your manuscript has been deemed suitable for publication in PLOS ONE. Congratulations! Your manuscript is now being handed over to our production team.

Kind regards, 

on behalf of

Dr. Md. Monirul Islam 

Academic Editor

PLOS ONE